# Inferences on Covariance Matrix with Block-wise Correlation Structure

## Abstract

Utilizing the sample moments of variable means within groups, we develop a novel closed-form estimator for blockwise correlation matrix of $p$ variables. When the block number and group memberships of the variables are known, we demonstrate the asymptotic normality of parameter estimators and establish the stochastic convergence rate of the estimated blockwise correlation matrix and corresponding estimated covariance matrix, under certain moment conditions. The method ensures positive semi-definiteness of the estimated covariance matrix without requiring a predetermined variable order, and can be applicable for high-dimensional data. Moreover, to estimate the number of blocks and recover their memberships, respectively, we employ the ridge-type ratio criterion and spectral clustering, and establish their consistency. Based on this, we extend the aforementioned properties of the asymptotic normality and stochastic convergence rate to the scenario where the group memberships are unknown and the block number is given. Extensive simulations and an empirical study of stock returns in the Chinese stock market are analyzed to illustrate the usefulness of our proposed methods.

## 1 Introduction

The covariance matrix plays a fundamental role in machine learning and multivariate analysis. Recently, with the emergence of high-dimensional data, modeling it has become a great challenge. The main reason is that the sample covariance matrix is often singular and irreversible when the matrix dimension surpasses the sample size (Kan & Zhou, 2007; Bai, 2008; Fan et al., 2011). To tackle the high-dimensional issue, one popular approach involves imposing a group or block structure on the correlation matrix of $p$-dimensional variables to achieve dimension reduction (Tsay & Pourahmadi, 2017; Archakov & Hansen, 2024). That is, the variables are clustered into $K$ ($K \ll p$) groups, and their correlation is determined based on their associated groups. This assumption is meaningful especially for finance data, in which the stock returns of companies in the same industries often exhibit similar correlations, while returns of firms across different industrial sectors tend to display weaker associations. The covariance matrix with blockwise correlation structure has gained significant popularity in various fields, including but not limited to: finance and risk management (Elton & Gruber, 1973; Engle & Kelly, 2012; Tsay & Pourahmadi, 2017; Millington & Niranjan, 2019), macroeconomics (Brownlees et al., 2022), resource assessment (Schuenemeyer & Gautier, 2010; Blondes et al., 2013), neuroscience and gene expression (Park et al., 2007; Wu & Smyth, 2012; Tan et al., 2015; Eisenach et al., 2020; Pircalabelu & Claeskens, 2020), and computer science (Zhang & Rao, 2013).

Even though the covariance matrix with blockwise correlation structure is widely used and well-motivated, its estimation poses challenges due to the inability of the maximum likelihood estimation (MLE) method to ensure positive semi-definiteness (Higham, 2002; Tsay & Pourahmadi, 2017). For illustration purpose, considering an $3 \times 3$ blockwise correlation matrix $\boldsymbol{R} = (\rho_{k_1 k_2})$ with $\rho_{12} = \rho_{13}$, the positive definite parameter domain for the correlation matrix $\boldsymbol{R}$ is defined as $\left\{ (\rho_{12}, \rho_{23}) | \rho_{23} > 2\rho_{12}^2 - 1 \right\}$. As noticed by Tsay & Pourahmadi (2017), as the values of $\rho_{12}$ and $\rho_{13}$ progressively approach the boundary of the parameter domain, the percentage of non-positive definiteness for the correlation matrix estimator based on MLE gradually increases. To address this issue, they proposed a method using the angle parameterization of its Cholesky factor. However, it is computationally complex, and the Cholesky decomposition necessitates a predetermined order of

variables, thereby limiting its applicability. Recently, Archakov & Hansen (2024) derived a canonical representation for the blockwise correlation matrix and dramatically simplified the evaluation of its maximum likelihood estimator, but it lacks rigorous theoretical justification. In addition, Engle & Kelly (2012) imposed the blockwise structure on a broad class of special correlation matrices, that is, the Dynamic Conditional Correlation models, and proved the asymptotic properties of their maximum likelihood estimators under Gaussian distribution and fixed $p$. Obviously, this method is also not universal in practice. Yang et al. (2024) proposed a closed-form covariance matrix estimator based on MLE by assuming a blockwise structure for the covariance matrix. Nevertheless, this assumption is stronger than that for a correlation matrix since homogeneous variance within each block is required. Moreover, due to errors involved in estimation of variance, the asymptotic distributions of parameters for the covariance matrix under the two structures are different.

In this paper, using the sample moments of variable means within groups, we propose a novel blockwise correlation matrix estimation method (BCME) in a closed form. When the block number and group memberships of variables are known, we derive the asymptotic normality of correlation coefficient estimators and establish the stochastic convergence rate of the estimated blockwise correlation and covariance matrix, under certain moment conditions. Compared to the Tsay & Pourahmadi (2017)'s method, our approach ensures the positive semi-definiteness of the covariance matrix estimation and holds the invariance of variable reordering. Furthermore, we employ the ridge-type ratio criterion and spectral clustering, to estimate the number of blocks and recover their memberships, respectively, and establish their consistency. Subsequently, for the scenario where the group memberships are unknown and the block number is given, the above properties of the asymptotic normality and stochastic convergence rate still hold. Various simulation studies and a real data analysis for portfolio allocation indicate that the proposed method outperforms the majority of existing methods.

The rest of the article is organized as follows. Section 2 describes the blockwise correlation matrix estimation and its asymptotic analysis when the block number and group memberships of variables are given. Secontion 3 introduces the block number determination, group membership recovery, and their consistency. Then, the same theoretical properties in Section 2 are extended to the scenario where the group memberships are unknown and the block number is known in Secontion 3. Section 4 and Section 5 present Monte Carlo studies and an empirical example, respectively. A brief discussion with some concluding remarks is given in Section 6. All technical details are relegated to the Appendix.

## 2 BLOCKWISE CORRELATION MATRIX ESTIMATION

### 2.1 BASIC NOTATIONS AND DEFINITION

Throughout the paper, vectors are denoted by lower-case bold letters, e.g., $\boldsymbol{\iota} = (\iota_1, \cdots, \iota_m)^\top \in \mathbb{R}^m$, and matrices by upper-case bold, e.g., $\boldsymbol{M} = (M_{ij}) \in \mathbb{R}^{m \times m}$. Define $\boldsymbol{0}_{m_1 \times m_2}$ and $\boldsymbol{1}_{m_1 \times m_2}$ as the $m_1 \times m_2$ vectors or matrices of all zeros and ones, respectively. Moreover, $\boldsymbol{0}_{m_1 \times 1}$ and $\boldsymbol{1}_{m_1 \times 1}$ are simplified as $\boldsymbol{0}_{m_1}$ and $\boldsymbol{1}_{m_1}$, respectively. Let $\boldsymbol{I}_m$ denote the identity matrix of dimension $m$. Here, $m$, $m_1$, and $m_2$ are any positive integers. In addition, let $\lambda_j(\boldsymbol{M})$ be the $j$-th largest eigenvalue of any generic matrix $\boldsymbol{M} \in \mathbb{R}^{m \times m}$ for $j = 1, \cdots, m$, $\|\boldsymbol{M}\|_F$ be the Frobenius norm of $\boldsymbol{M}$, $\|\boldsymbol{\iota}\|_v = (\sum_{j=1}^m |\iota_j|^v)^{1/v}$ be the vector $v$-norm or generalized matrix $v$-norm of generic vector $\boldsymbol{\iota} = (\iota_1, \cdots, \iota_m)^\top \in \mathbb{R}^m$ for $1 \le v \le \infty$, and induced norms be $\|\boldsymbol{M}\|_v = \sup_{\boldsymbol{\iota} \in \mathbb{R}^m : \boldsymbol{\iota} \ne \boldsymbol{0}_m} \frac{\|\boldsymbol{M}\boldsymbol{\iota}\|_v}{\|\boldsymbol{\iota}\|_v}$. The superscript $\top$ is the transpose of a vector or matrix. $\boldsymbol{1}_{\{\cdot\}}$ denotes an indicator function with condition in parentheses.

Next, we introduce the definition of the blockwise correlation matrix. Let $\mathbf{y}_i = (\mathbf{y}_{i1}, \cdots, \mathbf{y}_{ip})^\top \in \mathbb{R}^p$ be independent and identically distributed $p$-dimensional response vector with mean $\mathbb{E}(\mathbf{y}_i) = \boldsymbol{0}_p$ and covariance matrix $\mathrm{Cov}(\mathbf{y}_i) = \boldsymbol{\Sigma}$ for $i = 1, \cdots, n$. The covariance matrix $\boldsymbol{\Sigma}$ can be decomposed as $\boldsymbol{\Sigma} = \boldsymbol{\Lambda} \boldsymbol{R} \boldsymbol{\Lambda}$, where $\boldsymbol{R} = \mathrm{Corr}(\mathbf{y}_i)$ is the correlation matrix of $\mathbf{y}_i$ and $\boldsymbol{\Lambda} = \mathrm{diag}(\sigma_1, \cdots, \sigma_p)$ with $\sigma_j^2 = \mathrm{Var}(\mathbf{y}_{ij})$ for $i = 1, \cdots, n$ and $j = 1, \cdots, p$. We assume that the $p$-dimensional variables have a blockwise structure with $K$ groups. Specifically, denote $\mathbb{F} = \{1, \cdots, p\}$ as the full index set. For any given $K$, $\mathbb{F}$ is categorized into a total of $K$ groups as $\mathbb{F} = \bigcup_{k=1}^K \mathbb{S}_k$, where $\mathbb{S}_k$ collects the indices of variables within group $k$, $|\mathbb{S}_k| = p_k$, $\mathbb{S}_{k_1} \cap \mathbb{S}_{k_2} = \emptyset$, and $p = \sum_k p_k$, for $k_1 \ne k_2$

and $k, k_1, k_2 = 1, \cdots, K$. Here, for any set $\mathbb{M}$, $|\mathbb{M}|$ is the number of elements in $\mathbb{M}$. Then, without loss of generality, the elements of $\mathbf{y}_i$ in $\mathbb{F}$ have been sorted such that $\mathbf{y}_i = (\mathbf{y}_{i1}^\top, \cdots, \mathbf{y}_{iK}^\top)^\top$, where $\mathbf{y}_{ik} = (\mathrm{y}_{ij} : j \in \mathbb{S}_k) \in \mathbb{R}^{p_k}$ for any $k = 1, \cdots, K$. Moreover, $\boldsymbol{R}$ can be partitioned as $\boldsymbol{R} = (\boldsymbol{R}_{k_1 k_2})$ with $\boldsymbol{R}_{k_1 k_2} \in \mathbb{R}^{p_{k_1} \times p_{k_2}}$ (including the case $k_1 = k_2$), which is assumed to be blockwise, that is,

$$\boldsymbol{R}_{kk} = \rho_{kk}\mathbf{1}_{p_k \times p_k} + (1 - \rho_{kk})\boldsymbol{I}_{p_k} \text{ and } \boldsymbol{R}_{k_1 k_2} = \rho_{k_1 k_2}\mathbf{1}_{p_{k_1} \times p_{k_2}}, \tag{1}$$

where $\rho_{kk}, \rho_{k_1 k_2} \in [-1, 1]$, $\boldsymbol{R}_{k_1 k_2} = \boldsymbol{R}_{k_2 k_1}$. It is noteworthy that we only enforce the blockwise structure on the correlation matrix not on the covariance matrix, since the variances of variables are heterogeneous.

Based on this definition, we give some special notations used in this paper. Let $\boldsymbol{\Delta} := (\rho_{k_1 k_2}) \in \mathbb{R}^{K \times K}$ and $\boldsymbol{\rho} = \mathrm{vech}(\boldsymbol{\Delta}) \in \mathbb{R}^{K(K+1)/2}$ be the half-column-stacking vector of $\boldsymbol{\Delta}$. In addition, define $\boldsymbol{E}_k = \mathrm{diag}(\mathbf{0}_{p_1 \times p_1}, \cdots, \boldsymbol{I}_{p_k}, \cdots, \mathbf{0}_{p_K \times p_K}) \in \mathbb{R}^{p \times p}$ for any $k = 1, \cdots, K$. Let $\boldsymbol{\Theta} = (\boldsymbol{\theta}_1, \cdots, \boldsymbol{\theta}_p)^\top = (\Theta_{jk}) \in \mathbb{R}^{p \times K}$ be a membership matrix, where $\Theta_{jk} = 1$ if $j \in \mathbb{S}_k$ and $\Theta_{jk} = 0$ otherwise. Denote $\boldsymbol{D}_{k_1 k_2} = (D_{ij,k_1 k_2}) \in \mathbb{R}^{p \times p}$, where $D_{ij,k_1 k_2} = 1$ if $i \in \mathbb{S}_{k_1}$ and $j \in \mathbb{S}_{k_2}$, and $D_{ij,k_1 k_2} = 0$ otherwise.

## 2.2 BLOCKWISE CORRELATION MATRIX ESTIMATION

Following the definition of the blockwise correlation matrix, we define $\tilde{\mathbf{y}}_i = (\tilde{\mathbf{y}}_{i1}^\top, \cdots, \tilde{\mathbf{y}}_{iK}^\top)^\top := \boldsymbol{\Lambda}^{-1}\mathbf{y}_i$, then we obtain $\mathrm{Cov}(\tilde{\mathbf{y}}_i) = \mathrm{Corr}(\tilde{\mathbf{y}}_i) = \boldsymbol{R}$. Let $\tilde{\mathrm{z}}_{ik} = p_k^{-1}\mathbf{1}_{p_k}^\top\tilde{\mathbf{y}}_{ik}$ be the mean of variables within the $k$-th group for $k = 1, \cdots, K$. Simple calculation implies that $\mathrm{Var}(\tilde{\mathrm{z}}_{ik}) = \rho_{kk} + p_k^{-1}(1 - \rho_{kk})$ and $\mathrm{Cov}(\tilde{\mathrm{z}}_{ik_1}, \tilde{\mathrm{z}}_{ik_2}) = \rho_{k_1 k_2}$ for any $k_1 \neq k_2$. Subsequently, defining $\tilde{\mathbf{z}}_i = (\tilde{\mathrm{z}}_{i1}\mathbf{1}_{p_1}^\top, \cdots, \tilde{\mathrm{z}}_{iK}\mathbf{1}_{p_K}^\top)^\top \in \mathbb{R}^p$, the blockwise correlation matrix $\boldsymbol{R}$ can be decomposed as

$$\boldsymbol{R} = \boldsymbol{\Sigma}_{\tilde{\mathbf{z}}} + \boldsymbol{G}, \tag{2}$$

where $\boldsymbol{\Sigma}_{\tilde{\mathbf{z}}} = \mathrm{Cov}(\tilde{\mathbf{z}}_i)$ and $\boldsymbol{G} = \mathrm{diag}(\boldsymbol{G}_{11}, \cdots, \boldsymbol{G}_{KK})$ with $\boldsymbol{G}_{kk} = (1 - \rho_{kk})\boldsymbol{I}_{p_k} - p_k^{-1}(1 - \rho_{kk})\mathbf{1}_{p_k \times p_k}$. Therefore, to estimate $\boldsymbol{R}$, we can resort to the moments of $\tilde{\mathrm{z}}_{ik}$s.

Note that $\boldsymbol{\Lambda}$ is generally unknown in practice and needs to be estimated. Hence, we define $\hat{\boldsymbol{\Lambda}} = \mathrm{diag}(\hat{\sigma}_1, \cdots, \hat{\sigma}_p)$ as an estimator of $\boldsymbol{\Lambda}$ with $\hat{\sigma}_j^2 = n^{-1}\sum_{i=1}^n \mathrm{y}_{ij}^2$ for $j = 1, \cdots, p$. Replacing $\boldsymbol{\Lambda}$ with $\hat{\boldsymbol{\Lambda}}$, we obtain $\hat{\mathbf{y}}_i = (\hat{\mathbf{y}}_{i1}^\top, \cdots, \hat{\mathbf{y}}_{iK}^\top)^\top := \hat{\boldsymbol{\Lambda}}^{-1}\mathbf{y}_i$ and $\hat{\mathrm{z}}_{ik} = p_k^{-1}\mathbf{1}_{p_k}^\top\hat{\mathbf{y}}_{ik}$. Then, by (2), the blockwise correlation matrix estimation (BCME) for $\boldsymbol{R}$ are denoted as

$$\hat{\rho}_{kk} = \frac{\frac{p_k}{n}\sum_{i=1}^n \hat{\mathrm{z}}_{ik}\hat{\mathrm{z}}_{ik} - 1}{p_k - 1} = \frac{\frac{1}{n}\sum_{i=1}^n \mathbf{1}_{p_k}^\top\hat{\mathbf{y}}_{ik}\hat{\mathbf{y}}_{ik}^\top\mathbf{1}_{p_k} - p_k}{p_k(p_k - 1)},$$

$$\hat{\rho}_{k_1 k_2} = \frac{1}{n}\sum_{i=1}^n \hat{\mathrm{z}}_{ik_1}\hat{\mathrm{z}}_{ik_2} = \frac{\frac{1}{n}\sum_{i=1}^n \mathbf{1}_{p_{k_1}}^\top\hat{\mathbf{y}}_{ik_1}\hat{\mathbf{y}}_{ik_2}^\top\mathbf{1}_{p_{k_2}}}{p_{k_1}p_{k_2}}, \text{ for } k_1 > k_2. \tag{3}$$

Substituting (3) into (1), we obtain $\hat{\boldsymbol{R}}_{kk} = \hat{\rho}_{kk}\mathbf{1}_{p_k \times p_k} + (1 - \hat{\rho}_{kk})\boldsymbol{I}_{p_k}$ and $\hat{\boldsymbol{R}}_{k_1 k_2} = \hat{\rho}_{k_1 k_2}\mathbf{1}_{p_{k_1} \times p_{k_2}}$. Finally, the estimators of $\boldsymbol{\rho}$, $\boldsymbol{R}$, and $\boldsymbol{\Sigma}$ are represented as $\hat{\boldsymbol{\rho}} = (\hat{\rho}_{k_1 k_2}) \in \mathbb{R}^{K(K+1)/2}$, $\hat{\boldsymbol{R}} = (\hat{\boldsymbol{R}}_{k_1 k_2}) \in \mathbb{R}^{p \times p}$, and $\hat{\boldsymbol{\Sigma}} = \hat{\boldsymbol{\Lambda}}\hat{\boldsymbol{R}}\hat{\boldsymbol{\Lambda}}$, respectively. It is noteworthy that $\hat{\boldsymbol{R}}$ and $\hat{\boldsymbol{\Sigma}}$ are naturally positive semi-definite, since the eigenvalues of $\boldsymbol{G}$ is no less than 0.

## 2.3 ASYMPTOTIC ANALYSIS

To study the theoretical properties of out proposed method, we first assume the following three technical conditions.

**(C1)** (i) Write $\mathbf{y}_i := \boldsymbol{\Lambda}\boldsymbol{R}^{1/2}\boldsymbol{\epsilon}_i$ with $\boldsymbol{\epsilon}_i = (\epsilon_{i1}, \cdots, \epsilon_{ip})^\top \in \mathbb{R}^p$. We assume that $\epsilon_{ij}$s are independent and identically distributed (i.i.d.) with $\mathbb{E}(\epsilon_{ij}) = 0$ and $\mathrm{Var}(\epsilon_{ij}) = 1$, for any $i = 1, \cdots, n$ and $j = 1, \cdots, p$.
(ii) We assume $\sigma_j^2$ is bounded away from 0, for any $j = 1, \cdots, p$. In addition, there exist $\gamma_1 \in (0, 1]$ and $b_1 > 0$, such that for any $s_1 > 0$, $i = 1, \cdots, n$, and $j = 1, \cdots, p$, $P(|\mathrm{y}_{ij}| > s_1) \leq \exp(-(s_1/b_1)^{\gamma_1})$.

**(C2)** Assume that $2[\text{tr}(\boldsymbol{A}_{l_1}\boldsymbol{A}_{l_2})]_{K(K+1)/2 \times K(K+1)/2} + (\mu^{(4)} - 3)\boldsymbol{\Psi}^\top\boldsymbol{\Psi} \to \boldsymbol{Q}$ as $p \to \infty$ for a finite and positive definite matrix $\boldsymbol{Q}$, where $\boldsymbol{A}_l = \boldsymbol{R}^{1/2}\left\{ -\frac{\rho_{kk}(p_k-1)+1}{p_k(p_k-1)}\boldsymbol{E}_k + \frac{1}{p_k(p_k-1)}\boldsymbol{D}_{kk} \right\}\boldsymbol{R}^{1/2}$ when $l = k + (k-1)K - \sum_{k_3=0}^{k-1} k_3$, $\boldsymbol{A}_l = \boldsymbol{R}^{1/2}\left\{ \frac{1}{2p_{k_1}p_{k_2}}(\boldsymbol{D}_{k_1 k_2} + \boldsymbol{D}_{k_2 k_1}) - \frac{\rho_{k_1 k_2}}{2}(\frac{1}{p_{k_1}}\boldsymbol{E}_{k_1} + \frac{1}{p_{k_2}}\boldsymbol{E}_{k_2}) \right\}\boldsymbol{R}^{1/2}$ when $l = k_1 + (k_2-1)K - \sum_{k_3=0}^{k_2-1} k_3$ for $k_1 > k_2$ and $k, k_1, k_2 = 1, \cdots, K$, $\boldsymbol{\Psi}$ is defined in Lemma 1, and $\mu^{(4)} = \mathbb{E}(\epsilon_{ij}^4)$ for any $i = 1, \cdots, n$ and $j = 1, \cdots, p$.

**(C3)** Assume $p_k/p \to \pi_k \in (0, 1)$ for any $k = 1, \cdots, K$ as $p \to \infty$. In addition, $K$ is fixed.

Condition (C1)(i) introduces the moment conditions of $\boldsymbol{\epsilon}_i$, which has been widely used in related literature (Fan et al., 2011; Yamada et al., 2017; Feng et al., 2022; Zheng et al., 2022) and is weaker than the distributional assumption required in Tsay & Pourahmadi (2017) and Yang et al. (2024). Condition (C1)(ii) requires the distribution of $\mathbf{y}_i$ to have exponential-type tail, ensuring that the distribution does not have "heavy tails" (e.g., Cauchy distribution), which is necessary for the consistent estimation of the variance of $\mathbf{y}_{ij}$ (Fan et al., 2011; Feng et al., 2022). Condition (C2) is a standard assumption to ensure the covariance matrix of the estimated parameters converges a positive definite matrix. If Condition (C2) is invalid, then multicollinearity problems may arise. This is similar to the condition assumed in Zou et al. (2017). In addition, its rationality is shown in Appendix F. Condition (C3) indicates that the number of blocks is finite but the dimension of block submatrices is divergent as $p \to \infty$. This condition is also employed in Yamada et al. (2017). Based on the above three conditions, we obtain the asymptotic property of $\hat{\boldsymbol{\rho}}$ given below.

**Theorem 1** *Under Conditions (C1)-(C3), when $(\log p)^{6/\gamma_1 - 1} = o(n)$, as $\min\{n, p\} \to \infty$, we have that*

$$\sqrt{n}(\hat{\boldsymbol{\rho}} - \boldsymbol{\rho}) \xrightarrow{d} \mathcal{N}(\boldsymbol{0}_{K(K+1)/2}, \boldsymbol{Q}),$$

*where $\gamma_1$ and $\boldsymbol{Q}$ are defined in Conditions (C1) and (C2), respectively.*

Theorem 1 indicates that the convergence rate of $\hat{\boldsymbol{\rho}}$ is $\sqrt{n}$, which is independent of $p$. This results is reasonable since $\sigma_j$ for $j = 1, \cdots, p$ need to be estimated and involve estimation errors. To ensure the consistency of the estimators for $\sigma_j$s, the condition $(\log p)^{6/\gamma_1 - 1} = o(n)$ is required. Moreover, $\boldsymbol{Q}$ is unknown and needs to be estimated. By Condition (C2), $\hat{\boldsymbol{Q}}$ can be used as a consistent estimator of $\boldsymbol{Q}$ to make valid inferences, where $\hat{\boldsymbol{Q}}$ is calculated by replacing $\boldsymbol{\rho}$ in $\boldsymbol{Q}$ with $\hat{\boldsymbol{\rho}}$.

Based on the above Theorem 1, we next provide the stochastic convergence rate of the estimated blockwise correlation matrix $\hat{\boldsymbol{R}}$ and its related covariance matrix $\hat{\boldsymbol{\Sigma}}$.

**Theorem 2** *Under Conditions (C1)-(C3), when $(\log p)^{6/\gamma_1 - 1} = o(n)$, as $\min\{n, p\} \to \infty$, we have that*

$$p^{-1}\|\hat{\boldsymbol{R}} - \boldsymbol{R}\|_2 = O_p(n^{-1/2}), \quad p^{-1}\|\hat{\boldsymbol{R}} - \boldsymbol{R}\|_F = O_p(n^{-1/2}),$$

$$p^{-1}\|\hat{\boldsymbol{\Sigma}} - \boldsymbol{\Sigma}\|_2 = O_p(\sqrt{\frac{\log p}{n}}) \quad \text{and} \quad p^{-1}\|\hat{\boldsymbol{\Sigma}} - \boldsymbol{\Sigma}\|_F = O_p(\sqrt{\frac{\log p}{n}}).$$

## 3 BLOCK NUMBER DETERMINATION AND GROUP MEMBERSHIP RECOVERY

When true $K$ and $\mathbb{S}_k$s for all $k = 1, \cdots, K$ are given, by Theorem 1, the parameters of blockwise correlation matrix $\boldsymbol{R}$ can be estimated consistently. For a real-world application, however, the true $K$ and all $\mathbb{S}_k$s are unknown and need to be estimated correctly. Motivated by Lam & Yao (2012), Wang (2012), Ahn & Horenstein (2013), and Xia et al. (2015), we propose a ridge-type ratio (RR) to estimate $K$.

Before introducing the ridge-type ratio estimator for $K$, we present an additional condition and the bounds for the eigenvalues of $\boldsymbol{R}$ with the true block number $K$ as follows.

**(C4)** Assume that $c_1^{-1} < \lambda_K(\boldsymbol{\Delta}) \leq \cdots \leq \lambda_1(\boldsymbol{\Delta}) < c_1$ for a finite constant $c_1 > 0$.

**Proposition 1** *Under Conditions (C3) and (C4), as $p \to \infty$, we have that $c_{\lambda_1}^{-1}p \leq \lambda_K(\boldsymbol{R}) \leq \cdots \leq \lambda_1(\boldsymbol{R}) \leq c_{\lambda_1}p$ and $c_{\lambda_2}^{-1} \leq \lambda_p(\boldsymbol{R}) \leq \cdots \leq \lambda_{K+1}(\boldsymbol{R}) \leq c_{\lambda_2}$, for some finite constants $c_{\lambda_1}, c_{\lambda_2} > 0$.*

Condition (C4) assumes that $\boldsymbol{\Delta}$ is of full rank and Proposition 1 provides the order of eigenvalues of $\boldsymbol{R}$.

Then, the ridge-type ratio for estimating $K$ is denoted as

$$r_j = \frac{\lambda_j(\hat{\boldsymbol{R}}_{sam}) + \delta}{\lambda_{j+1}(\hat{\boldsymbol{R}}_{sam}) + \delta}, \quad j = 1, \cdots, p-1,$$

where $\hat{\boldsymbol{R}}_{sam} = n^{-1} \sum_{i=1}^{n} \hat{\mathbf{y}}_i \hat{\mathbf{y}}_i^\top$ with $\hat{\mathbf{y}}_i$ being defined in equation (3) and $\delta$ is a hyperparameter for ensuring that $\lambda_j(\hat{\boldsymbol{R}}_{sam}) + \delta > 0$ for any $j = 1, \cdots, p$. Consequently, the true number of blocks $K$ can be estimated by $\hat{K} = \arg\max_{j \in \{1, \cdots, p-1\}} r_j$. The consistency of $\hat{K}$ is ensured by the following theorem.

**Theorem 3** *Assume $\delta = o(p)$ and $p\sqrt{\log p/n} = o(\delta)$. Then, under Conditions (C1), (C3), and (C4), when $(\log p)^{6/\gamma_1 - 1} = o(n)$, as $\min\{n, p\} \to \infty$, we have that $P(\hat{K} = K) \to 1$, where $\gamma_1$ is defined in Condition (C1).*

Subsequently, to recover $\mathbb{S}_k$ for $k = 1, \cdots, K$, we estimate the membership matrix $\boldsymbol{\Theta}$ by clustering $p$ variables when the number of blocks is predetermined. After simple calculation, $\boldsymbol{R}$ can be re-expressed as $\boldsymbol{R} = \boldsymbol{\Theta}\boldsymbol{\Delta}\boldsymbol{\Theta}^\top + \boldsymbol{\Omega}$, where $\boldsymbol{\Omega}$ is a diagonal matrix to ensure the diagonal elements of $\boldsymbol{R}$ are 1s. Since the rank of $\boldsymbol{\Theta}\boldsymbol{\Delta}\boldsymbol{\Theta}^\top$ is $K$, we can rewrite $\boldsymbol{\Theta}\boldsymbol{\Delta}\boldsymbol{\Theta}^\top$ as $\boldsymbol{UVU}^\top$, where $\boldsymbol{V} \in \mathbb{R}^{K \times K}$ is a diagonal matrix consisting of the first $K$ largest eigenvalues of $\boldsymbol{\Theta}\boldsymbol{\Delta}\boldsymbol{\Theta}^\top$, and $\boldsymbol{U} \in \mathbb{R}^{p \times K}$ comprises the first $K$ eigenvectors of $\boldsymbol{\Theta}\boldsymbol{\Delta}\boldsymbol{\Theta}^\top$ as columns and has $K$ distinct rows. Therefore, we can resort to the row clustering of $\boldsymbol{U}$ to recover the blocks' memberships. Specifically, we eigen-decomposite $\hat{\boldsymbol{R}}_{sam}$ and take the first $K$ eigenvectors of $\hat{\boldsymbol{R}}_{sam}$ as the estimator of $\boldsymbol{U}$, denoted as $\hat{\boldsymbol{U}}$. Then, we can obtain the estimator of $\boldsymbol{\Theta}$, $\hat{\boldsymbol{\Theta}} = (\hat{\boldsymbol{\theta}}_1, \cdots, \hat{\boldsymbol{\theta}}_p)^\top$, by clustering the rows of $\hat{\boldsymbol{U}}$. Their almost sure convergence is proven in the Theorem II.3 of Su et al. (2019) under mild conditions, and demonstrated in Condition (C5).

**(C5)** Assume that for sufficiently large $n$ and $p$, $\sup_{1 \leq i \leq n} \sup_{1 \leq j \leq p} \mathbf{1}_{\{\hat{\boldsymbol{\theta}}_j \neq \boldsymbol{\theta}_j\}} = 0, a.s..$

Let $\hat{\boldsymbol{\rho}}_{\hat{\boldsymbol{\Theta}}}$ be an estimator of $\boldsymbol{\rho}$ with $\hat{\boldsymbol{\Theta}}$, we then get that

**Corollary 1** *Under Conditions (C1)-(C5), when $(\log p)^{6/\gamma_1 - 1} = o(n)$, as $\min\{n, p\} \to \infty$, we have that*

$$\sqrt{n}(\hat{\boldsymbol{\rho}}_{\hat{\boldsymbol{\Theta}}} - \boldsymbol{\rho}) \overset{d}{\longrightarrow} \mathcal{N}(\mathbf{0}_{K(K+1)/2}, \boldsymbol{\mathcal{Q}}),$$

*where $\gamma_1$ and $\boldsymbol{\mathcal{Q}}$ are defined in Conditions (C1) and (C2), respectively.*

The key of Corollary 1 is to prove $\hat{\boldsymbol{\rho}}_{\hat{\boldsymbol{\Theta}}} \overset{p}{\to} \hat{\boldsymbol{\rho}}$. It is straightforward with Condition (C5). For saving space, we are not reporting the proof. Hence, when $\mathbb{S}_k$s are unknown, according Corollary 1, the Theorem 2 still holds, which is also verified in simulation, see Table 8 in the Appendix G.

**Remark 1** *Recently, some researchers simultaneously estimate the model parameters and group memberships with given $K$ (Su et al., 2016; Liu et al., 2020; Zhu et al., 2023; Liu et al., 2024). However, their optimization functions are non-convex and require specific algorithms, which is theoretically complex and lacking generality. Hence, we propose the above two-step estimation (i.e., spectral clustering and BCME) to address this issue when $K$ is known.*

## 4 SIMULATION STUDIES

To evaluate the finite sample performance of our proposed method, we conduct Monte Carlo studies with the following setting. Specifically, for the blockwise correlation matrix $\boldsymbol{R}$ defined in (1), the off-diagonal elements of each diagonal block are set to $\rho_{kk} = 0.65 - 0.05(k-1)$, and elements of each off-diagonal block are set to $\rho_{k_1 k_2} = \rho_{k_1 k_1} - 0.25 - 0.05(k_2 - k_1 - 1)$ if $k_1 < k_2$ and $k_1$ is odd, and $\rho_{k_1 k_2} = \rho_{k_1 k_1} - 0.3 - 0.05(k_2 - k_1 - 1)$ if $k_1 < k_2$ and $k_1$ is even, respectively, for $k, k_1, k_2 = 1, \cdots, K$. This setting is similar to that in Wang (2012) and Zhao et al. (2022) and assures the resulting correlation matrix is positive definite for $K \leq 8$. Moreover, each block size is set to come from the sequence $(60, 90, 120, 150, 60, 90, 120, 150)$, that is, if $K = 1$, we set the

block size $p_1$ to be 60; if $K = 2$, we set two respective block sizes, $p_1$ and $p_2$, to be 60 and 90, and so forth, which is same as Saldana et al. (2017) and Hu et al. (2020). For $\mathbf{\Lambda}$, $\sigma_j$s are independently and identically generated from uniform distribution $\mathcal{U}(0, 1)$. In addition, to evaluate the robustness of our methods against a non-normal distribution, the response vector $\boldsymbol{y}_i$ is simulated by $\boldsymbol{y}_i = \mathbf{\Lambda} \boldsymbol{R}^{1/2} \boldsymbol{\varepsilon}_i$ with $\boldsymbol{\varepsilon}_i = (\varepsilon_{i1}, \cdots, \varepsilon_{ip})^\top \in \mathbb{R}^p$, where $\varepsilon_{ij}$s are independently and identically generated by three distributions: the standardized normal distribution $\mathcal{N}(0, 1)$, mixture normal distribution $0.9\mathcal{N}(0, 5/9) + 0.1\mathcal{N}(0, 5)$, and standardized exponential distribution. In simulation studies, all results are executed by 1,000 realizations with $n = 200, 500$, and $1,000$.

To verify the accuracy of the BCME method for estimating the blockwise correlation matrix and its associated covariance matrix, we assume that $K$ is known and $K \in \{2, 4, 6, 8\}$. In addition, we have included three additional competitors for the sake of comparison. They are the Tsay & Pourahmadi (2017, TP)'s estimator ($\hat{\boldsymbol{R}}_{TP}$ and $\hat{\boldsymbol{\Sigma}}_{TP}$), TP estimator with variable ordering ($\hat{\boldsymbol{R}}_{TP}^o$ and $\hat{\boldsymbol{\Sigma}}_{TP}^o$), and traditional QMLE estimator ($\hat{\boldsymbol{R}}_{QMLE}$ and $\hat{\boldsymbol{\Sigma}}_{QMLE}$). Then, we calculate the averages and standard deviations of two types of estimation errors, spectral-errors ($p^{-1}||\boldsymbol{M}_1 - \boldsymbol{R}||_2$ and $p^{-1}||\boldsymbol{M}_2 - \boldsymbol{\Sigma}||_2$) and Frobenius-errors ($p^{-1}||\boldsymbol{M}_1 - \boldsymbol{R}||_F$ and $p^{-1}||\boldsymbol{M}_2 - \boldsymbol{\Sigma}||_F$), across $\boldsymbol{M}_1 = \hat{\boldsymbol{R}}, \hat{\boldsymbol{R}}_{TP}, \hat{\boldsymbol{R}}_{TP}^o$, and $\hat{\boldsymbol{R}}_{QMLE}$ and $\boldsymbol{M}_2 = \hat{\boldsymbol{\Sigma}}, \hat{\boldsymbol{\Sigma}}_{TP}, \hat{\boldsymbol{\Sigma}}_{TP}^o$, and $\hat{\boldsymbol{\Sigma}}_{QMLE}$. Moreover, we report the proportion of positive semi-definiteness for estimated blockwise correlation matrix and its associated covariance matrix. The average execution time obtained through programming in Matlab using an Intel(R) Xeon(R) CPU (2.10 GHz) is also presented to reflect the computational complexity. Due to the high execution time intensity of the TP and QMLE approaches, we only report their results for $K = 2$, and 4. Table 1 illustrates three important findings when the elements of $\boldsymbol{\epsilon}_i$ follow $\mathcal{N}(0, 1)$. First, both the BCME and TP methods ensure positive semi-definiteness, whereas the QMLE approach cannot. Second, the BCME method dramatically reduces the execution time compared to the other two methods (0.037 sec v.s. 2957.841 sec and 8640.374 sec in $K = 4$ and $n = 1000$). Third, the BCME method addresses the requirement of a predetermined variable order in the TP method and achieves similar asymptotic efficiency for the TP method with variable ordering when $K = 2, 4$. Similarly, Tables 4 and 5 in the Appendix G yield analogous simulation results when $\boldsymbol{\epsilon}_i$ follows non-normal distributions.

Table 1: Comparison of the BCME estimators ($\hat{\boldsymbol{R}}, \hat{\boldsymbol{\Sigma}}$), TP estimators ($\hat{\boldsymbol{R}}_{TP}, \hat{\boldsymbol{\Sigma}}_{TP}$), TP estimators with variable ordering ($\hat{\boldsymbol{R}}_{TP}^o, \hat{\boldsymbol{\Sigma}}_{TP}^o$), and QMLE estimators ($\hat{\boldsymbol{R}}_{QMLE}, \hat{\boldsymbol{\Sigma}}_{QMLE}$) of the blockwise correlation matrix and corresponding covariance matrix when $\boldsymbol{\epsilon}_i$ follows a multivariate normal distribution $\mathcal{N}_p(\mathbf{0}_p, \boldsymbol{I}_p)$. AS and AF represent the averages of the spectral-error and Frobenius-error, respectively. SS and SF denote the standard deviations of the spectral-error and Frobenius-error, respectively. Pro. (%) is the proportion of positive semi-definiteness. Time (in seconds) is the average execution time.

| $(K,p)$ | | (2,150) | | | | (4,420) | | | | (6,570) | (8,840) |
|---|---|---|---|---|---|---|---|---|---|---|---|
| $n$ | Measures | $\Sigma(\hat{R})$ | $\Sigma_{TP}^o(R_{TP}^o)$ | $\Sigma_{TP}(\hat{R}_{TP})$ | $\Sigma_{QMLE}(\hat{R}_{QMLE})$ | $\Sigma(\hat{R})$ | $\Sigma_{TP}^o(R_{TP}^o)$ | $\Sigma_{TP}(\hat{R}_{TP})$ | $\Sigma_{QMLE}(\hat{R}_{QMLE})$ | $\Sigma(R)$ | $\Sigma(R)$ |
| 200 | AS | **0.019 (0.023)** | 0.019 (0.023) | 0.038 (0.108) | 0.026 (0.049) | **0.014 (0.025)** | 0.014 (0.026) | 0.027 (0.086) | 0.041 (0.124) | **0.014 (0.025)** | **0.010 (0.022)** |
| | SS | **0.009 (0.013)** | 0.009 (0.013) | 0.002 (0.000) | 0.028 (0.084) | **0.006 (0.012)** | 0.006 (0.012) | 0.002 (0.000) | 0.011 (0.037) | **0.006 (0.010)** | **0.004 (0.007)** |
| | AF | **0.020 (0.025)** | 0.020 (0.025) | 0.043 (0.111) | 0.028 (0.051) | **0.016 (0.030)** | 0.016 (0.030) | 0.040 (0.120) | 0.052 (0.158) | **0.016 (0.031)** | **0.013 (0.029)** |
| | SF | **0.008 (0.014)** | 0.008 (0.014) | 0.004 (0.004) | 0.029 (0.086) | **0.005 (0.012)** | 0.005 (0.012) | 0.002 (0.005) | 0.012 (0.004) | **0.005 (0.010)** | **0.003 (0.007)** |
| | Pro. | **100.0** | 100.0 | 100.0 | 94.3 | **100.0** | 100.0 | 100.0 | 10.1 | **100.0** | **100.0** |
| | Time | **0.004** | 12.987 | 20.370 | 30.918 | **0.023** | 2707.757 | 2819.685 | 1585.977 | **0.031** | **0.091** |
| 500 | AS | **0.012 (0.014)** | 0.012 (0.014) | 0.038 (0.108) | 0.014 (0.023) | **0.009 (0.016)** | 0.009 (0.016) | 0.026 (0.086) | 0.040 (0.125) | **0.009 (0.016)** | **0.006 (0.014)** |
| | SS | **0.006 (0.009)** | 0.006 (0.009) | 0.000 (0.000) | 0.017 (0.050) | **0.004 (0.007)** | 0.004 (0.007) | 0.000 (0.000) | 0.007 (0.024) | **0.003 (0.006)** | **0.002 (0.005)** |
| | AF | **0.013 (0.016)** | 0.013 (0.016) | 0.040 (0.110) | 0.015 (0.024) | **0.010 (0.019)** | 0.010 (0.019) | 0.038 (0.119) | 0.052 (0.163) | **0.010 (0.020)** | **0.008 (0.019)** |
| | SF | **0.005 (0.009)** | 0.005 (0.009) | 0.002 (0.002) | 0.018 (0.051) | **0.003 (0.007)** | 0.003 (0.007) | 0.001 (0.003) | 0.008 (0.026) | **0.003 (0.006)** | **0.002 (0.004)** |
| | Pro. | **100.0** | 100.0 | 100.0 | 98.0 | **100.0** | 100.0 | 100.0 | 1.8 | **100.0** | **100.0** |
| | Time | **0.004** | 13.339 | 22.053 | 73.623 | **0.028** | 2798.561 | 2900.917 | 4033.282 | **0.034** | **0.100** |
| 1000 | AS | **0.008 (0.010)** | 0.008 (0.010) | 0.038 (0.108) | 0.011 (0.017) | **0.006 (0.011)** | 0.006 (0.011) | 0.026 (0.086) | 0.040 (0.126) | **0.006 (0.011)** | **0.005 (0.010)** |
| | SS | **0.004 (0.006)** | 0.004 (0.006) | 0.000 (0.000) | 0.016 (0.046) | **0.003 (0.005)** | 0.003 (0.005) | 0.000 (0.000) | 0.006 (0.021) | **0.002 (0.004)** | **0.002 (0.003)** |
| | AF | **0.009 (0.011)** | 0.009 (0.011) | 0.039 (0.109) | 0.011 (0.018) | **0.007 (0.013)** | 0.007 (0.013) | 0.038 (0.118) | 0.053 (0.164) | **0.007 (0.014)** | **0.006 (0.013)** |
| | SF | **0.004 (0.007)** | 0.004 (0.007) | 0.001 (0.001) | 0.016 (0.047) | **0.002 (0.005)** | 0.002 (0.005) | 0.001 (0.003) | 0.007 (0.022) | **0.002 (0.004)** | **0.001 (0.003)** |
| | Pro. | **100.0** | 100.0 | 100.0 | 98.0 | **100.0** | 100.0 | 100.0 | 0.0 | **100.0** | **100.0** |
| | Time | **0.006** | 14.125 | 22.518 | 145.669 | **0.037** | 2957.841 | 3106.620 | 8640.374 | **0.042** | **0.117** |

We next study the finite sample performance of the RR method. To this end, we set $K \in \{2, 3, 4, 5, 6, 7, 8\}$ and $\delta = 10^{-2}pn^{-1/3}$. This choice of $\delta$ is similar to Xia et al. (2015) and Wang et al. (2022) and satisfies the theorem assumption defined in Theorem 3. In addition, we consider two measures to evaluate the performance of selection: (i) Mean: the mean of the estimated number of blocks $\hat{K}$, and (ii) CT: average percentage of the correct fit, $\mathbf{1}_{\{\hat{K}=K\}}$. Table 2 reports the Mean and CT for all $K$ when the entries of $\boldsymbol{\epsilon}_i$ follow $\mathcal{N}(0, 1)$. It shows that, the RR method completely restores the corresponding real block number when $p < n$. In addition, for $p > n$, the Mean of $\hat{K}$ is

gradually close to the real block number as the sample size $n$ increases while the CT rapidly tends to 1. Both of the results support Theorem 3. Similar findings can be observed when $\epsilon_i$ follows non-normal distributions (see Tables 6 and 7 in the Appendix G).

Table 2: Results of block number selection when $\epsilon_i$ follows a multivariate normal distribution $\mathcal{N}_p(\mathbf{0}_p, \mathbf{I}_p)$. CT is the average percentage of the correct fit. Mean is the mean of the estimated number of blocks.

|  | $n = 200$ | | $n = 500$ | | $n = 1000$ | |
|---|---|---|---|---|---|---|
|  | CT | Mean | CT | Mean | CT | Mean |
| $K = 2$ | 1.00 | 2.00 | 1.00 | 2.00 | 1.00 | 2.00 |
| $K = 3$ | 1.00 | 3.00 | 1.00 | 3.00 | 1.00 | 3.00 |
| $K = 4$ | 0.90 | 3.69 | 1.00 | 4.00 | 1.00 | 4.00 |
| $K = 5$ | 0.55 | 3.20 | 1.00 | 5.00 | 1.00 | 5.00 |
| $K = 6$ | 0.14 | 1.68 | 0.99 | 5.94 | 1.00 | 6.00 |
| $K = 7$ | 0.03 | 1.16 | 0.90 | 6.37 | 1.00 | 7.00 |
| $K = 8$ | 0.03 | 1.24 | 0.93 | 7.53 | 1.00 | 8.00 |

## 5 REAL DATA ANALYSIS

In this study, to demonstrate the superiority of our proposed methods, we analyze the daily returns of $p = 1076$ stocks belonging to the CSI Smallcap 500 Index and CSI 1000 Index from 2017 to 2021, where the data were collected from the WIND financial database.

We assess the performance of the out-of-sample portfolio by solving the Markowitz optimization problem (Markowitz, 1952). To this end, we estimate the covariance matrix $\mathbf{\Sigma}$ using the standard rolling window procedure with a window length of 12 quarters (Zivot & Wang, 2006; Zou et al., 2017). For each quarter $t$ ($t = 12, \cdots, 20$), we obtain the estimator $\hat{\mathbf{\Sigma}}_{BCME,t}^{RR} = \hat{\mathbf{\Lambda}}_t \hat{\mathbf{R}}_t \hat{\mathbf{\Lambda}}_t$ by employing the BCME method with the estimated block number $\hat{K}_t$ determined by the RR method. For the sake of comparison, we consider two additional BCME estimators ($\hat{\mathbf{\Sigma}}_{BCME,t}^{ind}$ and $\hat{\mathbf{\Sigma}}_{BCME,t}^{subind}$) constructed based on industries ($K_t = 16$) and sub-industries ($K_t = 64$) of stocks, respectively. In addition, we employ the Tsay & Pourahmadi (2017, TP)'s method with variable ordering under three different block numbers mentioned above ($\hat{\mathbf{\Sigma}}_{TP,t}^{o,RR}$, $\hat{\mathbf{\Sigma}}_{TP,t}^{o,ind}$, $\hat{\mathbf{\Sigma}}_{TP,t}^{o,subind}$). We also employ the methods of the Ledoit & Wolf (2004, LW1) ($\hat{\mathbf{\Sigma}}_{LW1,t}$), Ledoit & Wolf (2003, LW2) ($\hat{\mathbf{\Sigma}}_{LW2,t}$), Ledoit & Wolf (2020, LW3) ($\hat{\mathbf{\Sigma}}_{LW3,t}$), and Schäfer & Strimmer (2005, SS) ($\hat{\mathbf{\Sigma}}_{SS,t}$) to estimate the covariance matrix. Then, for each quarter $t$, we calculate 10 minimum variance portfolio weights by minimizing the portfolio variance, $\hat{\boldsymbol{\omega}}_t = \arg\min_{\boldsymbol{\omega} \in \mathbb{R}^p} \boldsymbol{\omega}^\top \mathbf{M}_t \boldsymbol{\omega}$, such that $\boldsymbol{\omega}^\top \mathbf{1}_p = 1$ and $\boldsymbol{\omega} \geq \mathbf{0}_p$, where $\mathbf{M}_t$ equals to the above 10 covariance matrix estimators. Next, let $\mathbf{Y}_t \in \mathbb{R}^{p \times T_t}$ denote the daily returns of stocks in quarter $t$, where $T_t$ is trading days at quarter $t$. Then, we compute the out-of-sample portfolios at quarter $t+1$ by $\mathbf{Y}_{t+1}^\top \hat{\boldsymbol{\omega}}_t$, across $\hat{\boldsymbol{\omega}}_t = \hat{\boldsymbol{\omega}}_{BCME,t}^{RR}, \hat{\boldsymbol{\omega}}_{BCME,t}^{ind}, \hat{\boldsymbol{\omega}}_{BCME,t}^{subind}, \hat{\boldsymbol{\omega}}_{TP,t}^{o,RR}, \hat{\boldsymbol{\omega}}_{TP,t}^{o,ind}, \hat{\boldsymbol{\omega}}_{TP,t}^{o,subind}, \hat{\boldsymbol{\omega}}_{LW1,t}, \hat{\boldsymbol{\omega}}_{LW2,t}, \hat{\boldsymbol{\omega}}_{LW3,t}, \hat{\boldsymbol{\omega}}_{SS,t}$, and $\hat{\boldsymbol{\omega}}_{Bench,t}$, where $\hat{\boldsymbol{\omega}}_{Bench,t}$ is the weight proportional to $t$-th quarter market capitalization and its corresponding out-of-sample portfolio is denoted as a benchmark.

To examine the out-of-sample portfolio performance (486 trading days from quarter 13 to quarter 20), we consider six commonly used measures: (i) the sample mean (Mean); (ii) sample standard deviation (SD); (iii) Sharpe ratio (SR); (iv) Turnover ratio (TR); (v) risk-adjusted excess return over the benchmark (Alpha); and (vi) Beta (the beta coefficient close to 1 indicates the out-of-sample portfolio has almost the same volatility as the benchmark). The results are provided in Table 3. Notably, due to the high execution time to obtain $\hat{\mathbf{\Sigma}}_{TP,t}^{o,ind}$ and $\hat{\mathbf{\Sigma}}_{TP,t}^{o,subind}$, we only calculate the out-of-sample portfolio based on $\hat{\mathbf{\Sigma}}_{TP,t}^{o,RR}$.

Table 3 indicates that the mean of the portfolio return based on BCME with RR is slightly larger than that of the portfolio return based on BCME with industries and sub-industries, as well as the portfolio return based on LW1, LW2, LW3, and SS methods, although these means are marginally smaller than that of the market portfolio return. In addition, the portfolio return based on BCME with RR ex-

hibits lower risk, measured by SD and Beta. As a result, the Sharpe ratio of the portfolio return based on BCME with RR is 30%, 20%, 10%,8% 5%, 5%, and 3% (e.g., $30\% = \{0.065 - 0.050\}/0.050$) higher than that of the portfolio return based on LW2, market portfolio return, portfolio return based LW1, BCME with industries, BCME with sub-industries, SS, and LW3, respectively. Finally, our method significantly outperforms the TP method with variable ordering in terms of execution time (0.036 sec v.s. 13415.565 sec), although both exhibit similar efficiency. This is particularly valuable in the ever-changing stock market. In sum, although the turnover ratio based on our method is not satisfactory, the block structure is significant for portfolio management and our proposed framework is highly effective for portfolio analysis.

Table 3: The sample mean (Mean), sample standard deviation (SD), Sharpe ratio (SR), Turnover ratio (TR), Alpha, and Beta calculated from 486 trading days of returns (%) in the market portfolio and portfolios constructed by BCME, TP, LW1, LW2, LW3, and SS methods, respectively, from 2020 to 2021 on the Chinese stock market, and the averaged execution time (Time, in seconds) to estimate corresponding covariance matrices for 9 quarters. The numbers within parentheses represent the standard errors of the alpha and beta coefficients, respectively. Dashes indicate null values or procedures that were not executed due to prohibitively time-intensity. The superscript $* * *$ denotes significance levels of 1%. Both $\uparrow$ and $\downarrow$ indicate better performance.

| | Mean ($\uparrow$) | SD ($\downarrow$) | SR ($\uparrow$) | TR ($\downarrow$) | Alpha ($\uparrow$) | Beta ($\downarrow$) | Time ($\downarrow$) |
|---|---|---|---|---|---|---|---|
| Market | **0.073** | 1.303 | 0.054 | **0.159** | 0 | 1 | - |
| BCME(RR) | **0.069** | **1.010** | **0.065** | 0.448 | **0.038 (0.038)** | **0.429*** (0.029)** | **0.036** |
| BCME(ind) | 0.062 | 0.965 | 0.060 | 0.308 | 0.030 (0.035) | 0.439*** (0.027) | 0.040 |
| BCME(subind) | 0.063 | 0.952 | 0.062 | 0.318 | 0.030 (0.034) | 0.451*** (0.026) | 0.038 |
| TP(RR) | 0.069 | 1.010 | 0.065 | 0.448 | 0.038 (0.038) | 0.429*** (0.029) | 13415.565 |
| TP(ind) | - | - | - | - | - | - | - |
| TP(subind) | - | - | - | - | - | - | - |
| LW1 | 0.058 | **0.921** | 0.059 | 0.287 | 0.020 (0.028) | 0.518*** (0.022) | 3.522 |
| LW2 | 0.050 | 0.922 | 0.050 | 0.286 | 0.017 (0.033) | 0.440*** (0.025) | 11.264 |
| LW3 | 0.066 | 0.990 | 0.063 | 0.276 | 0.022 (0.027) | 0.608*** (0.021) | 0.159 |
| SS | 0.061 | 0.930 | 0.062 | 0.277 | 0.022 (0.028) | 0.541*** (0.021) | 3.922 |

## 6 CONCLUSION AND REMARKS

We propose BCME to estimate a covariance matrix with blockwise correlation structure in high-dimensional settings. When the block number and group memberships of variables are known, the theoretical properties of the parameter estimators, the estimated blockwise correlation and covariance matrix are established under certain moment conditions. In addition, we utilize the ridge-type ratio criterion and spectral clustering to estimate the number of blocks and recover their memberships for a blockwise correlation matrix, and proved their consistency. Subsequently, we extend the properties of the asymptotic normality and stochastic convergence rate to the scenario where the group memberships are unknown and the block number is given. An application for analyzing portfolio returns in the Chinese stock market and simulation studies present superior performance of our proposed methods.

To expand the applicability of our proposed methods, we consider two major avenues for future research. First, extend the BCME, RR, and spectral clustering methods and establish their theoretical properties when $K$ is divergent, including $K/n \in (0, \infty]$ or $K/p \in (0, 1]$. This is reasonable and common in ultra high-dimensional data. Second, develop general methods for estimating a quantiled moment and choosing the number of blocks when the quantiled moment has a block structure. These extensions would further reveal the usefulness of our proposed methods for inferences on structured blockwise moment.

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

## APPENDIX

## A  THREE TECHNICAL LEMMAS

**Lemma 1** *Let $\boldsymbol{\epsilon}_i = (\epsilon_{i1}, \cdots, \epsilon_{ip})^\top \in \mathbb{R}^p$ for $i = 1, \cdots, n$ be independent and identically distributed random vectors, and satisfies Condition (C1). Define*

$$\mathbf{q}_{n,p} = \frac{1}{n} \sum_{i=1}^{n} \begin{pmatrix} \mathrm{vec}^\top(\boldsymbol{A}_1) \\ \vdots \\ \mathrm{vec}^\top(\boldsymbol{A}_L) \end{pmatrix} \mathrm{vec}(\boldsymbol{\epsilon}_i \boldsymbol{\epsilon}_i^\top - \boldsymbol{I}_p),$$

*where $\boldsymbol{A}_l = (A_{j_1 j_2}^{(l)}) \in \mathbb{R}^{p \times p}$ is symmetric for $l = 1, \cdots, L$ with $L < \infty$. Then, we have $\mathbb{E}(\mathbf{q}_{n,p}) = \boldsymbol{0}_L$, and*

$$\mathrm{Cov}(\mathbf{q}_{n,p}) = 2n^{-1}[\mathrm{tr}(\boldsymbol{A}_{l_1} \boldsymbol{A}_{l_2})]_{L \times L} + (\mu^{(4)} - 3)n^{-1} \boldsymbol{\Psi}^\top \boldsymbol{\Psi}, \tag{4}$$

*where $\boldsymbol{\Psi} = (\boldsymbol{\psi}_1, \cdots, \boldsymbol{\psi}_L) \in \mathbb{R}^{p \times L}$ with $\boldsymbol{\psi}_l := (A_{11}^{(l)}, \cdots, A_{pp}^{(l)})^\top \in \mathbb{R}^p$ for $l = 1, \cdots, L$. If there exists a positive definite matrix $\boldsymbol{\mathcal{Q}} \in \mathbb{R}^{L \times L}$ such that $n\mathrm{Cov}(\mathbf{q}_{n,p}) \to \boldsymbol{\mathcal{Q}}$, then, we have*

$$n^{1/2} \mathbf{q}_{n,p} \xrightarrow{d} \mathcal{N}(0, \boldsymbol{\mathcal{Q}}). \tag{5}$$

**Proof.** The equation (4) is directly exacted from Chen et al. (2010). To prove equation (5), by Cramér-Wold device, it suffices to establish the asymptotic normality of $\boldsymbol{\xi}^\top \mathbf{q}_{n,p}$ for arbitrary vector $\boldsymbol{\xi} = (\xi_1, \cdots, \xi_L)^\top > \boldsymbol{0}_L \in \mathbb{R}^L$. Denote $\mathrm{q}_{\xi i} = \sum_l \xi_l \mathrm{vec}^\top(\boldsymbol{A}_l) \mathrm{vec}(\boldsymbol{\epsilon}_i \boldsymbol{\epsilon}_i^\top - \boldsymbol{I}_p)$. We have $\mathbb{E}(\mathrm{q}_{\xi i}) = 0$, $\mathrm{Var}(\mathrm{q}_{\xi i}) = \boldsymbol{\xi}^\top \{2[\mathrm{tr}(\boldsymbol{A}_{l_1} \boldsymbol{A}_{l_2})]_{L \times L} + (\mu^{(4)} - 3)\boldsymbol{\Psi}^\top \boldsymbol{\Psi}\}\boldsymbol{\xi}$, according to (4), and $\boldsymbol{\xi}^\top \mathbf{q}_{n,p} = n^{-1} \sum_{i=1}^{n} \mathrm{q}_{\xi i}$. To prove the asymptotic normality of $\boldsymbol{\xi}^\top \mathbf{q}_{n,p}$, it suffices to verify that the Lindeberg condition holds, that is,

$$\lim_{p \to \infty} \lim_{n \to \infty} \frac{1}{\sum_{i=1}^{n} \mathrm{Var}(\mathrm{q}_{\xi i})} \sum_{i=1}^{n} \int_{x^2 > c_\xi^2 \sum_{i=1}^{n} \mathrm{var}(\mathrm{q}_{\xi i})} x^2 dF_{\mathrm{q}_{\xi i}}(x) = 0, \tag{6}$$

where $F_{\mathrm{q}_{\xi i}}(x)$ is the cumulative distribution function of $\mathrm{q}_{\xi i}$ and $c_\xi$ is an arbitrary constant. Since $\mathrm{q}_{\xi i}$s are i.i.d., we have

$$\lim_{n \to \infty} \frac{1}{\sum_{i=1}^{n} \mathrm{var}(\mathrm{q}_{\xi i})} \sum_{i=1}^{n} \int_{x^2 > c_\xi^2 \sum_{i=1}^{n} \mathrm{var}(\mathrm{q}_{\xi i})} x^2 dF_{\mathrm{q}_{\xi i}}(x)$$

$$= \lim_{n \to \infty} \frac{1}{\mathrm{var}(\mathrm{q}_{\xi i})} \int_{x^2 > c_\xi^2 \sum_{i=1}^{n} \mathrm{var}(\mathrm{q}_{\xi i})} x^2 dF_{\mathrm{q}_{\xi i}}(x) = 0,$$

where the last equality is due to that the variance of $\mathrm{q}_{\xi i}$ exists and finite. Thus, (6) holds, which completes the entire proof of this lemma.

**Lemma 2** *Under Conditions (C1) and (C3), we have $\|n^{-1} \sum_{i=1}^{n} \tilde{\mathbf{y}}_i \tilde{\mathbf{y}}_i^\top - \boldsymbol{R}\|_F = O_p(p/\sqrt{n})$.*

**Proof.** We evaluate the expectation of $\|n^{-1}\sum_{i=1}^{n}\tilde{\mathbf{y}}_i\tilde{\mathbf{y}}_i^{\top} - \mathbf{R}\|_F^2$ to prove the lemma. We obtain

$$\mathbb{E}\|n^{-1}\sum_{i=1}^{n}\tilde{\mathbf{y}}_i\tilde{\mathbf{y}}_i^{\top} - \mathbf{R}\|_F^2 = \mathbb{E}\|n^{-1}\sum_{i=1}^{n}\tilde{\mathbf{y}}_i\tilde{\mathbf{y}}_i^{\top}\|_F^2 - \|\mathbf{R}\|_F^2,$$

where $\mathbb{E}\|n^{-1}\sum_{i=1}^{n}\tilde{\mathbf{y}}_i\tilde{\mathbf{y}}_i^{\top}\|_F^2 = n^{-1}\mathbb{E}\|\tilde{\mathbf{y}}_i\tilde{\mathbf{y}}_i^{\top}\|_F^2 + (n-1)n^{-1}\mathbb{E}\{\mathrm{tr}(\tilde{\mathbf{y}}_{i_1}\tilde{\mathbf{y}}_{i_1}^{\top}\tilde{\mathbf{y}}_{i_2}\tilde{\mathbf{y}}_{i_2}^{\top})\}$ for $i, i_1, i_2 = 1, \cdots, n$. According to Lemma 1, we have

$$\mathbb{E}\|\tilde{\mathbf{y}}_i\tilde{\mathbf{y}}_i^{\top}\|_F^2 = 2\mathrm{tr}(\mathbf{R}^2) + (\mu^{(4)}-3)\mathrm{tr}(\mathbf{R}\circ\mathbf{R}) + \mathrm{tr}^2(\mathbf{R}),$$
$$\mathbb{E}\{\mathrm{tr}(\tilde{\mathbf{y}}_{i_1}\tilde{\mathbf{y}}_{i_1}^{\top}\tilde{\mathbf{y}}_{i_2}\tilde{\mathbf{y}}_{i_2}^{\top})\} = \mathrm{tr}(\mathbf{R}^2).$$

This, together with the Condition (C3), implies

$$\mathbb{E}\|n^{-1}\sum_{i=1}^{n}\tilde{\mathbf{y}}_i\tilde{\mathbf{y}}_i^{\top} - R\|_F^2 = n^{-1}\{\mathrm{tr}(\mathbf{R}^2) + p(\mu^{(4)}-3) + \mathrm{tr}^2(\mathbf{R})\}$$

$$\leq n^{-1}\Big\{p^2\Big[\sum_{k\in\{1,\cdots,K\}}\pi_k^2 + 2\sum_{\substack{k_1>k_2 \\ k_1,k_2\in\{1,\cdots,K\}}}\pi_{k_1}\pi_{k_2}\Big] + p(\mu^{(4)}-3) + p^2\Big\}$$

$$= O(p^2/n),$$

which completes the proof.

**Lemma 3** *Under Condition (C1) and $(\log p)^{6/\gamma_1-1} = o(n)$, as $\min\{n,p\}\to\infty$, we have that*

$$\max_{j\in\{1,\cdots,p\}}|\hat{\sigma}_j - \sigma_j| = O_p\Big(\sqrt{\frac{\log p}{n}}\Big).$$

**Proof.** By Lemma A.2 of Fan et al. (2011) and formula (1.3) of Merlevède et al. (2011), together with Condition (C1) and $(\log p)^{6/\gamma_1-1} = o(n)$, we have that

$$P\Big\{\max_{j\in\{1,\cdots,p\}}|\hat{\sigma}_j^2 - \sigma_j^2| \geq c_\sigma\sqrt{\frac{\log p}{n}}\Big\}\to 0$$

as $\min\{n,p\}\to\infty$, for a finite constant $c_\sigma > 0$. Then, $\max_{j\in\{1,\cdots,p\}}|\hat{\sigma}_j - \sigma_j||\hat{\sigma}_j + \sigma_j| = O_p\{(\log p/n)^{1/2}\}$ and $\max_{j\in\{1,\cdots,p\}}|\hat{\sigma}_j - \sigma_j| = O_p\{(\log p/n)^{1/2}\}$, which completes the entire proof of this lemma.

# B    PROOF OF PROPOSITION 1

Recall that $\mathbf{R} = \mathbf{\Theta}\mathbf{\Delta}\mathbf{\Theta}^{\top} + \mathbf{\Omega}$, where $\mathbf{\Omega} = \mathrm{diag}((1-\rho_{11})\mathbf{I}_{p_1},\cdots,(1-\rho_{KK})\mathbf{I}_{p_K})$. Since the rank of $\mathbf{\Theta}\mathbf{\Delta}\mathbf{\Theta}^{\top}$ is $K$, we know that the last $p-K$ eigenvalues of $\mathbf{R}$ are positive and finite. Then, we consider the $K$ eigenvalues of $\mathbf{\Theta}\mathbf{\Delta}\mathbf{\Theta}^{\top}$ to give the property of the first $K$ eigenvalues of $\mathbf{R}$. Defining $\mathbf{P} = \mathrm{diag}(\sqrt{p_1},\cdots,\sqrt{p_K}) \in \mathbb{R}^{K\times K}$, we have $\mathbf{\Theta}\mathbf{\Delta}\mathbf{\Theta}^{\top} = \mathbf{\Theta}\mathbf{P}^{-1}\mathbf{P}\mathbf{\Delta}\mathbf{P}\mathbf{P}^{-1}\mathbf{\Theta}^{\top}$, where $\mathbf{\Theta}\mathbf{P}^{-1}$ is orthonormal. Let $\mathbf{U}_\Delta\mathbf{V}_\Delta\mathbf{U}_\Delta^{\top}$ be the eigendecomposition of $\mathbf{P}\mathbf{\Delta}\mathbf{P}$. Then the eigenvector matrix $\mathbf{U}$ of $\mathbf{\Theta}\mathbf{\Delta}\mathbf{\Theta}^{\top}$ is equal to $\mathbf{\Theta}\mathbf{P}^{-1}\mathbf{U}_\Delta$. The eigenvalues of $\mathbf{\Theta}\mathbf{\Delta}\mathbf{\Theta}^{\top}$ are equal to those of $\mathbf{P}\mathbf{\Delta}\mathbf{P}$. Under Conditions (C3) and (C4), we have $c_{\lambda_1}^{-1}p \leq p\lambda_K(\mathbf{\Delta}) \leq \lambda_k(\mathbf{P}\mathbf{\Delta}\mathbf{P}) \leq p\lambda_1(\mathbf{\Delta}) \leq c_{\lambda_1}p$, for $k\in[K]$ and some constant $c_{\lambda_1} > 0$, which completes the proof.

# C    PROOF OF THEOREM 1

We will prove this theorem in two steps via the Delta method. In step I, we prove that $\hat{\boldsymbol{\rho}}$ can be approximated by its first order Taylor expansion. In step II, we demonstrate that $\hat{\boldsymbol{\rho}}$ is asymptotical normal.

**Step I.** Define $\mathbf{\Lambda}_k = \mathrm{diag}(\sigma_{1k},\cdots,\sigma_{p_kk}) := \mathrm{diag}(\sigma_j, j\in\mathbb{S}_k) \in \mathbb{R}^{p_k\times p_k}$, $\hat{\mathbf{\Lambda}}_k = \mathrm{diag}(\hat{\sigma}_{1k},\cdots,\hat{\sigma}_{p_kk}) := \mathrm{diag}(\hat{\sigma}_j, j\in\mathbb{S}_k) \in \mathbb{R}^{p_k\times p_k}$, and $\mathbf{\Pi}_{k_1k_2} = n^{-1}\sum_{i=1}^{n}\tilde{\mathbf{y}}_{ik_1}\tilde{\mathbf{y}}_{ik_2}^{\top} \in \mathbb{R}^{p_k\times p_k}$.

Then, formula (2.2) of the main paper can be rewritten as

$$\hat{\rho}_{k_1 k_2} = \frac{\mathbf{1}_{p_{k_1}}^\top \hat{\mathbf{\Lambda}}_{k_1}^{-1} \mathbf{\Lambda}_{k_1} \mathbf{\Pi}_{k_1 k_2} \mathbf{\Lambda}_{k_2} \hat{\mathbf{\Lambda}}_{k_2}^{-1} \mathbf{1}_{p_k}}{p_{k_1} p_{k_2}}, \text{ for } k_1 > k_2,$$

$$\hat{\rho}_{kk} = \frac{\mathbf{1}_{p_k}^\top \hat{\mathbf{\Lambda}}_k^{-1} \mathbf{\Lambda}_k \mathbf{\Pi}_{kk} \mathbf{\Lambda}_k \hat{\mathbf{\Lambda}}_k^{-1} \mathbf{1}_{p_k} - p_k}{p_k(p_k - 1)}.$$

We treat $\hat{\rho}_{k_1 k_2}$ for $k_1 > k_2$ as a function of $\hat{\sigma}_j^2$ for $j \in \mathbb{S}_{k_1} \cup \mathbb{S}_{k_2}$ and $\mathbf{\Pi}_{k_1 k_2}$. Employing Taylor series expansion, we have that

$$\begin{aligned}
\hat{\rho}_{k_1 k_2} - \rho_{k_1 k_2} = &-\frac{\rho_{k_1 k_2}}{2} \sum_{c \in \{k_1, k_2\}} \Big\{ p_c^{-1} \sum_{j=1}^{p_c} \sigma_{jc}^{-2} (\hat{\sigma}_{jc}^2 - \sigma_{jc}^2) \Big\} \\
&+ (p_{k_1} p_{k_2})^{-1} \text{vec}^\top (\mathbf{1}_{p_{k_1} \times p_{k_2}}) \text{vec}(\mathbf{\Pi}_{k_1 k_2} - \mathbf{R}_{k_1 k_2}) \\
&+ \frac{3\rho_{k_1 k_2}}{8} \sum_{c \in \{k_1, k_2\}} \Big\{ p_c^{-1} \sum_{j=1}^{p_c} \sigma_{jc}^{*-5} \sigma_{jc} (\hat{\sigma}_{jc}^2 - \sigma_{jc}^2)^2 \Big\} \\
&+ \frac{\rho_{k_1 k_2}}{8 p_{k_1} p_{k_2}} \sum_{j_1=1}^{p_{k_1}} \sum_{j_2=1}^{p_{k_2}} \Big\{ \sigma_{j_1 k_1}^{*-3} \sigma_{j_1 k_1} \sigma_{j_2 k_2}^{*-3} \sigma_{j_2 k_2} (\hat{\sigma}_{j_1 k_1}^2 - \sigma_{j_1 k_1}^2)(\hat{\sigma}_{j_2 k_2}^2 - \sigma_{j_2 k_2}^2) \Big\} \\
&- \frac{1}{p_{k_1} p_{k_2}} \sum_{c \in \{k_1, k_2\}} \Big\{ \text{vec}^\top (\mathbf{\Pi}_{k_1 k_2} - \rho_{k_1 k_2} \mathbf{1}_{p_{k_1} \times p_{k_2}}) \mathbf{\Xi}_c (\hat{\sigma}_{1c}^2 - \sigma_{1c}^2, \cdots, \hat{\sigma}_{p_c c}^2 - \sigma_{p_c c}^2)^\top \Big\} \\
= &H_{11} + H_{12} + I_1 + I_2 + I_3,
\end{aligned}$$

where $\sigma_{jc}^*$ is between $\sigma_{jc}$ and $\hat{\sigma}_{jc}$, $\mathbf{\Xi}_c = \mathbf{1}_{p_{\tilde{c}}} \otimes \text{diag}(\sigma_{1c}^{*-3} \sigma_{1c}, \cdots, \sigma_{p_c c}^{*-3} \sigma_{1c})$, $\tilde{c} \in \{k_1, k_2\}$, and $\tilde{c} \neq c$. Here, $\otimes$ is the Kronecker product. To study the asymptotic property of $\hat{\rho}_{k_1 k_2}$, we first prove that $I_1$, $I_2$, and $I_3$ are $o_p(n^{-1/2})$. According to Lemma 3, $\sigma_{jc}$ and $\hat{\sigma}_{jc}$ are bounded with probability tending to 1.

For $I_1$, we have that

$$\sqrt{n}|I_1| \leq C_1 \sqrt{n} \max_{c \in \{k_1, k_2\}} p_c^{-1} \sum_{j=1}^{p_c} (\hat{\sigma}_{jc}^2 - \sigma_{jc}^2)^2 \leq C_1 n^{-1/2} \log p \to 0,$$

for some finite positive constant $C_1$. For $I_2$, we have that

$$\sqrt{n}|I_2| \leq C_2 \sqrt{n} (p_{k_1} p_{k_2})^{-1} \sum_{j_1=1}^{p_{k_1}} \sum_{j_2=1}^{p_{k_2}} (\hat{\sigma}_{j_1 k_1}^2 - \sigma_{j_1 k_1}^2)(\hat{\sigma}_{j_2 k_2}^2 - \sigma_{j_2 k_2}^2) \leq C_2 n^{-1/2} \log p \to 0,$$

for some finite positive constant $C_2$. For $I_3$, by the Cauchy–Schwarz inequality, we have that

$$\begin{aligned}
\sqrt{n}|I_3| \leq &C_3 (p_{k_1} p_{k_2})^{-1} \| \mathbf{\Pi}_{k_1 k_2} - \rho_{k_1 k_2} \mathbf{1}_{p_{k_1} \times p_{k_2}} \|_F \max_{c \in \{k_1, k_2\}} \| \mathbf{\Xi}_c \|_F \sqrt{\log p} \\
\leq &C_3 (p_{k_1} p_{k_2})^{-1} O_p(p/\sqrt{n}) \sqrt{\log p} \max_{c \in \{k_1, k_2\}} \| \mathbf{\Xi}_c \|_F = O_p \{(\log p/n)^{1/2}\} \to 0,
\end{aligned}$$

for some finite positive constant $C_3$. Therefore, $\hat{\rho}_{k_1 k_2} - \rho_{k_1 k_2} = H_{11} + H_{12} + o_p(n^{-1/2})$ for $k_1 > k_2$.

Employing similar techniques, we obtain that

$$\begin{aligned}
\hat{\rho}_{kk} - \rho_{kk} = &-\frac{\rho_{kk}(p_k - 1) + 1}{p_k(p_k - 1)} \sum_{j=1}^{p_k} \big\{ \sigma_{jk}^{-2}(\hat{\sigma}_{jk}^2 - \sigma_{jk}^2) \big\} \\
&+ \frac{1}{p_k(p_k - 1)} \text{vec}^\top (\mathbf{1}_{p_k \times p_k}) \text{vec}(\mathbf{\Pi}_{kk} - \mathbf{R}_{kk}) + o_p(n^{-1/2}) \\
= &H_{21} + H_{22} + o_p(n^{-1/2}),
\end{aligned}$$

which completes the first part of the proof.

**Step II.** In this step, we study the asymptotic distribution of $\hat{\boldsymbol{\rho}}$ according to Lemma 1. We have that

$$H_{11} = \text{vec}^\top \Big( -\frac{\rho_{k_1 k_2}}{2} \boldsymbol{R}^{1/2} (\frac{1}{p_{k_1}} \boldsymbol{E}_{k_1} + \frac{1}{p_{k_2}} \boldsymbol{E}_{k_2}) \boldsymbol{R}^{1/2} \Big) \frac{1}{n} \sum_{i=1}^n \text{vec}(\boldsymbol{\epsilon}_i \boldsymbol{\epsilon}_i^\top - \boldsymbol{I}_p),$$

$$H_{12} = \text{vec}^\top \Big( \frac{1}{2 p_{k_1} p_{k_2}} \boldsymbol{R}^{1/2} (\boldsymbol{D}_{k_1 k_2} + \boldsymbol{D}_{k_2 k_1}) \boldsymbol{R}^{1/2} \Big) \frac{1}{n} \sum_{i=1}^n \text{vec}(\boldsymbol{\epsilon}_i \boldsymbol{\epsilon}_i^\top - \boldsymbol{I}_p),$$

$$H_{21} = \text{vec}^\top \Big( -\frac{\rho_{kk}(p_k - 1) + 1}{p_k(p_k - 1)} \boldsymbol{R}^{1/2} \boldsymbol{E}_k \boldsymbol{R}^{1/2} \Big) \frac{1}{n} \sum_{i=1}^n \text{vec}(\boldsymbol{\epsilon}_i \boldsymbol{\epsilon}_i^\top - \boldsymbol{I}_p),$$

$$H_{22} = \text{vec}^\top \Big( \frac{1}{p_k(p_k - 1)} \boldsymbol{R}^{1/2} \boldsymbol{D}_{kk} \boldsymbol{R}^{1/2} \Big) \frac{1}{n} \sum_{i=1}^n \text{vec}(\boldsymbol{\epsilon}_i \boldsymbol{\epsilon}_i^\top - \boldsymbol{I}_p).$$

Then, $\hat{\boldsymbol{\rho}} - \boldsymbol{\rho}$ can be rewritten as

$$\tilde{\mathbf{q}}_{n,p} := \hat{\boldsymbol{\rho}} - \boldsymbol{\rho} = \frac{1}{n} \sum_{i=1}^n \begin{pmatrix} \text{vec}^\top(\boldsymbol{A}_1) \\ \vdots \\ \text{vec}^\top(\boldsymbol{A}_L) \end{pmatrix} \text{vec}(\boldsymbol{\epsilon}_i \boldsymbol{\epsilon}_i^\top - \boldsymbol{I}_p) + o_p(n^{-1/2}),$$

where $L = \frac{K(K+1)}{2} < \infty$, $\boldsymbol{A}_l = \boldsymbol{R}^{1/2} \Big\{ -\frac{\rho_{kk}(p_k-1)+1}{p_k(p_k-1)} \boldsymbol{E}_k + \frac{1}{p_k(p_k-1)} \boldsymbol{D}_{kk} \Big\} \boldsymbol{R}^{1/2}$ when $l = k + (k-1)K - \sum_{k_3=0}^{k-1} k_3$, and $\boldsymbol{A}_l = \boldsymbol{R}^{1/2} \Big\{ \frac{1}{2 p_{k_1} p_{k_2}} (\boldsymbol{D}_{k_1 k_2} + \boldsymbol{D}_{k_2 k_1}) - \frac{\rho_{k_1 k_2}}{2} (\frac{1}{p_{k_1}} \boldsymbol{E}_{k_1} + \frac{1}{p_{k_2}} \boldsymbol{E}_{k_2}) \Big\} \boldsymbol{R}^{1/2}$ when $l = k_1 + (k_2 - 1)K - \sum_{k_3=0}^{k_2-1} k_3$ for $k_1 > k_2$ and $k, k_1, k_2 = 1, \cdots, K$.

According to Lemma 1, we have $\text{Cov}(\tilde{\mathbf{q}}_{n,p}) = 2n^{-1} [\text{tr}(\boldsymbol{A}_{l_1} \boldsymbol{A}_{l_2})]_{L \times L} + (\mu^{(4)} - 3) n^{-1} \boldsymbol{\Psi}^\top \boldsymbol{\Psi}$. According to Condition (C2), $n \text{Cov}(\tilde{\mathbf{q}}_{n,p}) \to \boldsymbol{\mathcal{Q}}$. Thus, by Lemma 1, we have $n^{1/2}(\hat{\boldsymbol{\rho}} - \boldsymbol{\rho}) \xrightarrow{d} \mathcal{N}(\boldsymbol{0}_{K(K+1)/2}, \boldsymbol{\mathcal{Q}})$, which completes the entire proof.

## D  PROOF OF THEOREM 2

By Theorem 1, we can obtain $\|\hat{\boldsymbol{\rho}} - \boldsymbol{\rho}\|_2 = O_p(n^{-1/2})$. This, together with Lemma 3 and Condition (C3), imply that

$$\|\hat{\boldsymbol{R}} - \boldsymbol{R}\|_F = \Big[ \sum_{\substack{k \in \{1, \cdots, K\}}} p_k(p_k - 1)(\hat{\rho}_{kk} - \rho_{kk})^2 + 2 \sum_{\substack{k_1 > k_2 \\ k_1, k_2 \in \{1, \cdots, K\}}} p_{k_1} p_{k_2} (\hat{\rho}_{k_1 k_2} - \rho_{k_1 k_2})^2 \Big]^{1/2}$$

$$\leq \Big\{ p^2 O_p(n^{-1}) \Big[ \sum_{\substack{k \in \{1, \cdots, K\}}} \pi_k^2 + 2 \sum_{\substack{k_1 > k_2 \\ k_1, k_2 \in \{1, \cdots, K\}}} \pi_{k_1} \pi_{k_2} \Big] \Big\}^{1/2} = O_p(\frac{p}{\sqrt{n}}).$$

Analogously, we obtain

$$\|\hat{\boldsymbol{\Sigma}} - \boldsymbol{\Sigma}\|_F = \Big\{ \sum_{\substack{k \in \{1, \cdots, K\}}} \sum_{\substack{j \in \{1, \cdots, p_k\}}} (\hat{\sigma}_{jk}^2 - \sigma_{jk}^2)^2$$

$$+ 2 \sum_{\substack{k \in \{1, \cdots, K\}}} \sum_{\substack{j_1 > j_2 \\ j_1, j_2 \in \{1, \cdots, p_k\}}} [\hat{\sigma}_{j_1 k} \hat{\sigma}_{j_2 k} (\hat{\rho}_{kk} - \rho_{kk}) + (\hat{\sigma}_{j_1 k} \hat{\sigma}_{j_2 k} - \sigma_{j_1 k} \sigma_{j_2 k}) \rho_{kk}]^2$$

$$+ 2 \sum_{\substack{k_1 > k_2 \\ k_1, k_2 \in \{1, \cdots, K\}}} \sum_{\substack{j_1 \in \{1, \cdots, p_{k_1}\}}} \sum_{\substack{j_2 \in \{1, \cdots, p_{k_2}\}}} [\hat{\sigma}_{j_1 k_1} \hat{\sigma}_{j_2 k_2} (\hat{\rho}_{k_1 k_2} - \rho_{k_1 k_2}) + (\hat{\sigma}_{j_1 k_1} \hat{\sigma}_{j_2 k_2} - \sigma_{j_1 k_1} \sigma_{j_2 k_2}) \rho_{k_1 k_2}]^2 \Big\}^{1/2}$$

$$\leq \Big\{ O_p(\frac{\log p}{n}) \Big[ p + p^2 \sum_{\substack{k \in \{1, \cdots, K\}}} \pi_k^2 + 2p^2 \sum_{\substack{k_1 > k_2 \\ k_1, k_2 \in \{1, \cdots, K\}}} \pi_{k_1} \pi_{k_2} \Big] \Big\}^{1/2} = O_p(p \sqrt{\frac{\log p}{n}}).$$

Subsequently, by 4.67(a) in Seber (2008, p. 68), we have $\|\hat{\boldsymbol{R}} - \boldsymbol{R}\|_2 \leq \|\hat{\boldsymbol{R}} - \boldsymbol{R}\|_F = O_p(\frac{p}{\sqrt{n}})$ and $\|\hat{\boldsymbol{\Sigma}} - \boldsymbol{\Sigma}\|_2 \leq \|\hat{\boldsymbol{\Sigma}} - \boldsymbol{\Sigma}\|_F = O_p(p\sqrt{\frac{\log p}{n}})$, which completes the entire proof.

## E    PROOF OF THEOREM 3

To prove this theorem, we consider the following two steps. In Step I, we prove $\lambda_k(\hat{\boldsymbol{R}}_{sam}) + \delta = O_p(p)$ for $k \leq K$, and $\lambda_k(\hat{\boldsymbol{R}}_{sam}) + \delta = O_p(1 \vee \delta)$ for $k \geq K + 1$. In Step II, we derive the consistency of $\hat{K}$. Here, $m_1 \vee m_2 = \max\{m_1, m_2\}$ for any $m_1$ and $m_2$.

**STEP I.** By the definition of $\hat{\boldsymbol{R}}_{sam}$ and triangle inequality, we have that

$$
\begin{aligned}
\|\hat{\boldsymbol{R}}_{sam} - \boldsymbol{R}\|_F &= \|\hat{\boldsymbol{\Lambda}}^{-1}\boldsymbol{\Lambda}\frac{1}{n}\sum_{i=1}^{n}\tilde{\mathbf{y}}_i\tilde{\mathbf{y}}_i^{\top}\boldsymbol{\Lambda}\hat{\boldsymbol{\Lambda}}^{-1} - \boldsymbol{R}\|_F \\
&= \|\hat{\boldsymbol{\Lambda}}^{-1}\boldsymbol{\Lambda}(\frac{1}{n}\sum_{i=1}^{n}\tilde{\mathbf{y}}_i\tilde{\mathbf{y}}_i^{\top} - \boldsymbol{R})\boldsymbol{\Lambda}\hat{\boldsymbol{\Lambda}}^{-1} + \hat{\boldsymbol{\Lambda}}^{-1}\boldsymbol{\Lambda}\boldsymbol{R}(\boldsymbol{\Lambda}\hat{\boldsymbol{\Lambda}}^{-1} - \boldsymbol{I}_p) + (\hat{\boldsymbol{\Lambda}}^{-1}\boldsymbol{\Lambda} - \boldsymbol{I}_p)\boldsymbol{R}\|_F \\
&\leq \|\hat{\boldsymbol{\Lambda}}^{-1}\boldsymbol{\Lambda}\|_2^2\frac{1}{n}\sum_{i=1}^{n}\tilde{\mathbf{y}}_i\tilde{\mathbf{y}}_i^{\top} - \boldsymbol{R}\|_F + \|\hat{\boldsymbol{\Lambda}}^{-1}\boldsymbol{\Lambda}\|_2\|\boldsymbol{R}\|_F\|\boldsymbol{\Lambda}\hat{\boldsymbol{\Lambda}}^{-1} - \boldsymbol{I}_p\|_2 \\
&\quad + \|\hat{\boldsymbol{\Lambda}}^{-1}\boldsymbol{\Lambda} - \boldsymbol{I}_p\|_2\|\boldsymbol{R}\|_F \\
&= O_p(\|\frac{1}{n}\sum_{i=1}^{n}\tilde{\mathbf{y}}_i\tilde{\mathbf{y}}_i^{\top} - \boldsymbol{R}\|_F + \|\hat{\boldsymbol{\Lambda}}^{-1}\boldsymbol{\Lambda} - \boldsymbol{I}_p\|_2\|\boldsymbol{R}\|_F).
\end{aligned}
$$

According to Lemmas 2 and 3, we have that $\|n^{-1}\sum_{i=1}^{n}\tilde{\mathbf{y}}_i\tilde{\mathbf{y}}_i^{\top} - \boldsymbol{R}\|_F = O_p(p/\sqrt{n})$, $\|\hat{\boldsymbol{\Lambda}}^{-1}\boldsymbol{\Lambda} - \boldsymbol{I}_p\|_2 = O_p(\sqrt{\log p/n})$, and $\|\boldsymbol{R}\|_F = O(p)$. Thus, we obtain $\|\hat{\boldsymbol{R}}_{sam} - \boldsymbol{R}\|_F = O_p(p\sqrt{\log p/n})$ and $\|\hat{\boldsymbol{R}}_{sam} - \boldsymbol{R}\|_2 = O_p(p\sqrt{\log p/n})$.

Let $\delta = o(p)$ and $p\sqrt{\log p/n} = o(\delta)$. Then, by the Weyl's inequality, we have

$$
\lambda_k(\boldsymbol{R}) - \|\hat{\boldsymbol{R}}_{sam} - \boldsymbol{R}\|_2 + \delta \leq \lambda_k(\hat{\boldsymbol{R}}_{sam}) + \delta \leq \lambda_k(\boldsymbol{R}) + \|\hat{\boldsymbol{R}}_{sam} - \boldsymbol{R}\|_2 + \delta,
$$

for $k = 1, \cdots, p$. This, together with Proposition 1, implies $\lambda_k(\hat{\boldsymbol{R}}_{sam}) + \delta = O_p(p)$ for $k \leq K$, and $\lambda_k(\hat{\boldsymbol{R}}_{sam}) + \delta = O_p(1 \vee \delta)$ for $k > K$.

**STEP II.** As $n$ and $p$ are sufficiently large, we can get

$$
\max_{j<K} r_j = \max_{j<K} \frac{\lambda_j(\hat{\boldsymbol{R}}_{sam}) + \delta}{\lambda_{j+1}(\hat{\boldsymbol{R}}_{sam}) + \delta} \leq \frac{\lambda_1(\hat{\boldsymbol{R}}_{sam}) + \delta}{\lambda_K(\hat{\boldsymbol{R}}_{sam}) + \delta} = O_p(1),
$$

and

$$
\max_{j>K} r_j = \max_{j>K} \frac{\lambda_j(\hat{\boldsymbol{R}}_{sam}) + \delta}{\lambda_{j+1}(\hat{\boldsymbol{R}}_{sam}) + \delta} \leq \frac{\lambda_{K+1}(\hat{\boldsymbol{R}}_{sam}) + \delta}{\lambda_p(\hat{\boldsymbol{R}}_{sam}) + \delta} = O_p(1).
$$

Then, similarly, as long as $n$ and $p$ are sufficiently large, we obtain

$$
\frac{\lambda_K(\hat{\boldsymbol{R}}_{sam}) + \delta}{\lambda_{K+1}(\hat{\boldsymbol{R}}_{sam}) + \delta} = O_p(p \wedge \delta^{-1}p),
$$

which diverges in probability towards infinity. Here, $m_1 \wedge m_2 = \min\{m_1, m_2\}$ for any $m_1$ and $m_2$. This completes the last step and the entire proof.

## F    RATIONALITY OF CONDITION (C2)

For illustration purpose, we set $\mu^{(4)} = 3$. We next give the concrete form of $\operatorname{tr}(\boldsymbol{A}_{l_1}\boldsymbol{A}_{l_2})$ for any $l_1, l_2 = 1, \cdots, K(K+1)/2$ in the following three cases.

**Case I.** Denote $\boldsymbol{Q}_k = -\frac{\rho_{kk}(p_k-1)+1}{p_k(p_k-1)}\boldsymbol{I}_{p_k} + \frac{1}{p_k(p_k-1)}\mathbf{1}_{p_k \times p_k}$. For $l_1 = k_1 + (k_1-1)K - \sum_{h=0}^{k_1-1}h$ and $l_2 = k_2 + (k_2-1)K - \sum_{h=0}^{k_2-1}h$, we obtain

$$\text{tr}(\boldsymbol{A}_{l_1}\boldsymbol{A}_{l_2}) = \text{tr}(\boldsymbol{R}_{k_2 k_1}\boldsymbol{Q}_{k_1}\boldsymbol{R}_{k_1 k_2}\boldsymbol{Q}_{k_2}) \to \rho_{k_1 k_2}^2(1-\rho_{k_1 k_1})(1-\rho_{k_2 k_2}),$$

for $k_1, k_2 = 1, \cdots, K$.

**Case II.** Since $\text{tr}(\boldsymbol{A}_{l_1}\boldsymbol{A}_{l_2}) = \text{tr}(\boldsymbol{A}_{l_2}\boldsymbol{A}_{l_1})$, we only present the results of $l_1 = k+(k-1)K-\sum_{h=0}^{k-1}h$ and $l_2 = k_1 + (k_2-1)K - \sum_{h=0}^{k_2-1}h$. After algebraic calculation, we obtain

$$\text{tr}(\boldsymbol{A}_{l_1}\boldsymbol{A}_{l_2}) = \frac{1}{2p_{k_1}p_{k_2}}\text{tr}(\boldsymbol{Q}_k\boldsymbol{R}_{kk_1}\mathbf{1}_{k_1 \times k_2}\boldsymbol{R}_{k_2 k} + \boldsymbol{Q}_k\boldsymbol{R}_{kk_2}\mathbf{1}_{k_2 \times k_1}\boldsymbol{R}_{k_1 k})$$

$$- \frac{\rho_{k_1 k_2}}{2}\text{tr}(\frac{1}{p_{k_1}}\boldsymbol{Q}_k\boldsymbol{R}_{kk_1}\boldsymbol{R}_{k_1 k} + \frac{1}{p_{k_2}}\boldsymbol{Q}_k\boldsymbol{R}_{kk_2}\boldsymbol{R}_{k_2 k})$$

$$\to (1-\rho_{kk})\{\rho_{kk_1}\rho_{kk_2} - \frac{1}{2}\rho_{kk_1}^2\rho_{k_1 k_2} - \frac{1}{2}\rho_{kk_2}^2\rho_{k_1 k_2}\},$$

for $k, k_1, k_2 = 1, \cdots, K$ and $k_1 > k_2$.

**Case III.** For $l_1 = k_1 + (k_2-1)K - \sum_{h=0}^{k_2-1}h$ and $l_2 = k_3 + (k_4-1)K - \sum_{h=0}^{k_4-1}h$, we obtain

$$\text{tr}(\boldsymbol{A}_{l_1}\boldsymbol{A}_{l_2}) = \frac{1}{4p_{k_1}p_{k_2}p_{k_3}p_{k_4}}\text{tr}(\boldsymbol{R}^{1/2}(\boldsymbol{D}_{k_1 k_2} + \boldsymbol{D}_{k_2 k_1})\boldsymbol{R}(\boldsymbol{D}_{k_3 k_4} + \boldsymbol{D}_{k_4 k_3})\boldsymbol{R}^{1/2})$$

$$- \frac{\rho_{k_3 k_4}}{4p_{k_1}p_{k_2}}\text{tr}(\boldsymbol{R}^{1/2}(\boldsymbol{D}_{k_1 k_2} + \boldsymbol{D}_{k_2 k_1})\boldsymbol{R}(\frac{1}{p_{k_3}}\boldsymbol{E}_{k_3} + \frac{1}{p_{k_4}}\boldsymbol{E}_{k_4})\boldsymbol{R}^{1/2})$$

$$- \frac{\rho_{k_1 k_2}}{4p_{k_3}p_{k_4}}\text{tr}(\boldsymbol{R}^{1/2}(\boldsymbol{D}_{k_3 k_4} + \boldsymbol{D}_{k_4 k_3})\boldsymbol{R}(\frac{1}{p_{k_1}}\boldsymbol{E}_{k_1} + \frac{1}{p_{k_2}}\boldsymbol{E}_{k_2})\boldsymbol{R}^{1/2})$$

$$+ \frac{\rho_{k_1 k_2}\rho_{k_3 k_4}}{4}\text{tr}(\boldsymbol{R}^{1/2}(\frac{1}{p_{k_1}}\boldsymbol{E}_{k_1} + \frac{1}{p_{k_2}}\boldsymbol{E}_{k_2})\boldsymbol{R}(\frac{1}{p_{k_3}}\boldsymbol{E}_{k_3} + \frac{1}{p_{k_4}}\boldsymbol{E}_{k_4})\boldsymbol{R}^{1/2})$$

$$= B_1 + B_2 + B_3 + B_4,$$

where $k_1, k_2, k_3, k_4 = 1, \cdots, K$, $k_1 > k_2$, and $k_3 > k_4$.

After algebraic calculation, we obtain

$$B_1 = \frac{1}{4p_{k_1}p_{k_2}p_{k_3}p_{k_4}}\Big\{\text{tr}(\mathbf{1}_{k_1 \times k_2}\boldsymbol{R}_{k_2 k_3}\mathbf{1}_{k_3 \times k_4}\boldsymbol{R}_{k_4 k_1}) + \text{tr}(\mathbf{1}_{k_2 \times k_1}\boldsymbol{R}_{k_1 k_3}\mathbf{1}_{k_3 \times k_4}\boldsymbol{R}_{k_4 k_2})$$

$$+ \text{tr}(\mathbf{1}_{k_1 \times k_2}\boldsymbol{R}_{k_2 k_4}\mathbf{1}_{k_4 \times k_3}\boldsymbol{R}_{k_3 k_1}) + \text{tr}(\mathbf{1}_{k_2 \times k_1}\boldsymbol{R}_{k_1 k_4}\mathbf{1}_{k_4 \times k_3}\boldsymbol{R}_{k_3 k_2})\Big\}$$

$$\to \frac{1}{2}(\rho_{k_2 k_3}\rho_{k_1 k_4} + \rho_{k_1 k_3}\rho_{k_2 k_4}).$$

Analogously, we obtain

$$B_2 = -\frac{\rho_{k_3 k_4}}{4p_{k_1}p_{k_2}}\Big\{\text{tr}(\frac{1}{p_{k_3}}\mathbf{1}_{k_1 \times k_2}\boldsymbol{R}_{k_2 k_3}\boldsymbol{R}_{k_3 k_1}) + \text{tr}(\frac{1}{p_{k_4}}\mathbf{1}_{k_1 \times k_2}\boldsymbol{R}_{k_2 k_4}\boldsymbol{R}_{k_4 k_1})$$

$$+ \text{tr}(\frac{1}{p_{k_3}}\mathbf{1}_{k_2 \times k_1}\boldsymbol{R}_{k_1 k_3}\boldsymbol{R}_{k_3 k_2}) + \text{tr}(\frac{1}{p_{k_4}}\mathbf{1}_{k_2 \times k_1}\boldsymbol{R}_{k_1 k_4}\boldsymbol{R}_{k_4 k_2})\Big\}$$

$$\to -\frac{\rho_{k_3 k_4}}{2}(\rho_{k_2 k_3}\rho_{k_1 k_3} + \rho_{k_2 k_4}\rho_{k_1 k_4}),$$

and $B_3 \to -\frac{\rho_{k_1 k_2}}{2}(\rho_{k_1 k_3}\rho_{k_1 k_4} + \rho_{k_2 k_4}\rho_{k_2 k_3})$. For $B_4$, we have

$$B_4 = \frac{\rho_{k_1 k_2}\rho_{k_3 k_4}}{4}\Big\{\text{tr}(\frac{1}{p_{k_1}p_{k_3}}\boldsymbol{R}_{k_1 k_3}\boldsymbol{R}_{k_3 k_1}) + \text{tr}(\frac{1}{p_{k_2}p_{k_3}}\boldsymbol{R}_{k_2 k_3}\boldsymbol{R}_{k_3 k_2})$$

$$+ \text{tr}(\frac{1}{p_{k_1}p_{k_4}}\boldsymbol{R}_{k_1 k_4}\boldsymbol{R}_{k_4 k_1}) + \text{tr}(\frac{1}{p_{k_2}p_{k_4}}\boldsymbol{R}_{k_2 k_4}\boldsymbol{R}_{k_4 k_2})\Big\}$$

$$\to \frac{\rho_{k_1 k_2}\rho_{k_3 k_4}}{4}(\rho_{k_1 k_3}^2 + \rho_{k_2 k_3}^2 + \rho_{k_1 k_4}^2 + \rho_{k_2 k_4}^2).$$

Combining the above results, we immediately know that $B_1 + B_2 + B_3 + B_4$ is convergent.

Since every elements in $[\text{tr}(\boldsymbol{A}_{l_1}\boldsymbol{A}_{l_2})]_{K(K+1)/2 \times K(K+1)/2}$ are convergent and $K < \infty$, $[\text{tr}(\boldsymbol{A}_{l_1}\boldsymbol{A}_{l_2})]$ is also convergent, which implies that our Condition (C2) is sensible.

# G  ADDITIONAL SIMULATION RESULTS

In this section, we present two different types of additional simulation studies. First, the simulation settings are the same as those in Section 4, except that the elements of $\epsilon_i$ are i.i.d. from the mixture normal distribution $0.9\mathcal{N}(0, 5/9) + 0.1\mathcal{N}(0, 5)$ and standardized exponential distribution. We find that the results yield similar patterns to those in Tables 1 and 2, which demonstrates the robustness of the BCME and RR methods, shown in Tables 4–7. Second, we demonstrate the results of the BCME estimations with given $K$ when the group memberships are unknown and $\epsilon_i$ follows a multivariate normal distribution $\mathcal{N}_p(\mathbf{0}_p, \mathbf{I}_p)$ in Table 8, which verifies the corollary 1. In addition, the similar results are yielded when $\epsilon_i$ follows non-normal distributions, but they are not reported here to save space.

Table 4: Comparison of the BCME estimators ($\hat{\mathbf{R}}$, $\hat{\mathbf{\Sigma}}$), TP estimators ($\hat{\mathbf{R}}_{TP}$, $\hat{\mathbf{\Sigma}}_{TP}$), TP estimators with variable ordering ($\hat{\mathbf{R}}_{TP}^o$, $\hat{\mathbf{\Sigma}}_{TP}^o$), and QMLE estimators ($\hat{\mathbf{R}}_{QMLE}$, $\hat{\mathbf{\Sigma}}_{QMLE}$) of the blockwise correlation matrix and corresponding covariance matrix when the elements of $\epsilon_i$ follow a mixture normal distribution $0.9\mathcal{N}(0, 5/9) + 0.1\mathcal{N}(0, 5)$. AS and AF represent the averages of the spectral-error and Frobenius-error, respectively. SS and SF denote the standard deviations of the spectral-error and Frobenius-error, respectively. Pro. (%) is the proportion of positive semi-definiteness. Time (in seconds) is the average execution time.

| $(K, p)$ | | (2,150) | | | | (4,420) | | | | (6,570) | (8,840) |
|---|---|---|---|---|---|---|---|---|---|---|---|
| $n$ | Measures | $\hat{\mathbf{\Sigma}}(\hat{\mathbf{R}})$ | $\hat{\mathbf{\Sigma}}_{TP}^o(\hat{\mathbf{R}}_{TP}^o)$ | $\hat{\mathbf{\Sigma}}_{TP}(\hat{\mathbf{R}}_{TP})$ | $\hat{\mathbf{\Sigma}}_{QMLE}(\hat{\mathbf{R}}_{QMLE})$ | $\hat{\mathbf{\Sigma}}(\hat{\mathbf{R}})$ | $\hat{\mathbf{\Sigma}}_{TP}^o(\hat{\mathbf{R}}_{TP}^o)$ | $\hat{\mathbf{\Sigma}}_{TP}(\hat{\mathbf{R}}_{TP})$ | $\hat{\mathbf{\Sigma}}_{QMLE}(\hat{\mathbf{R}}_{QMLE})$ | $\hat{\mathbf{\Sigma}}(\hat{\mathbf{R}})$ | $\hat{\mathbf{\Sigma}}(\hat{\mathbf{R}})$ |
| | AS | **0.021 (0.024)** | 0.021 (0.024) | 0.036 (0.108) | 0.029 (0.055) | **0.016 (0.026)** | 0.016 (0.026) | 0.027 (0.086) | 0.041 (0.123) | **0.014 (0.025)** | **0.012 (0.022)** |
| | SS | **0.008 (0.014)** | 0.008 (0.014) | 0.002 (0.000) | 0.028 (0.091) | **0.006 (0.011)** | 0.006 (0.011) | 0.002 (0.000) | 0.011 (0.039) | **0.005 (0.010)** | **0.004 (0.007)** |
| 200 | AF | **0.023 (0.026)** | 0.023 (0.026) | 0.042 (0.112) | 0.032 (0.058) | **0.018 (0.030)** | 0.018 (0.030) | 0.041 (0.120) | 0.052 (0.157) | **0.017 (0.031)** | **0.015 (0.029)** |
| | SF | **0.007 (0.014)** | 0.007 (0.014) | 0.004 (0.004) | 0.030 (0.093) | **0.005 (0.011)** | 0.005 (0.011) | 0.002 (0.005) | 0.013 (0.047) | **0.004 (0.009)** | **0.003 (0.007)** |
| | Pro. | **100.0** | 100.0 | 100.0 | 93.0 | **100.0** | 100.0 | 100.0 | 10.4 | **100.0** | **100.0** |
| | Time | **0.003** | 12.899 | 20.123 | 31.161 | **0.023** | 2681.941 | 2742.259 | 1585.685 | **0.031** | **0.092** |
| | AS | **0.013 (0.015)** | 0.013 (0.015) | 0.035 (0.108) | 0.019 (0.035) | **0.010 (0.016)** | 0.010 (0.016) | 0.026 (0.086) | 0.040 (0.125) | **0.009 (0.015)** | **0.008 (0.014)** |
| | SS | **0.005 (0.009)** | 0.005 (0.009) | 0.003 (0.007) | 0.024 (0.076) | **0.004 (0.007)** | 0.004 (0.007) | 0.001 (0.000) | 0.005 (0.014) | **0.003 (0.006)** | **0.002 (0.005)** |
| 500 | AF | **0.014 (0.017)** | 0.014 (0.017) | 0.038 (0.110) | 0.020 (0.037) | **0.012 (0.019)** | 0.012 (0.019) | 0.039 (0.118) | 0.053 (0.163) | **0.011 (0.019)** | **0.010 (0.019)** |
| | SF | **0.005 (0.010)** | 0.005 (0.010) | 0.003 (0.008) | 0.024 (0.077) | **0.003 (0.007)** | 0.003 (0.007) | 0.001 (0.003) | 0.005 (0.015) | **0.002 (0.006)** | **0.002 (0.004)** |
| | Pro. | **100.0** | 100.0 | 100.0 | 95.0 | **100.0** | 100.0 | 100.0 | 0.7 | **100.0** | **100.0** |
| | Time | **0.004** | 13.333 | 21.319 | 75.666 | **0.028** | 2781.171 | 2896.381 | 4001.367 | **0.035** | **0.102** |
| | AS | **0.009 (0.011)** | 0.009 (0.011) | 0.035 (0.108) | 0.012 (0.019) | **0.007 (0.011)** | 0.007 (0.011) | 0.026 (0.086) | 0.040 (0.125) | **0.006 (0.011)** | **0.005 (0.010)** |
| | SS | **0.004 (0.006)** | 0.004 (0.006) | 0.000 (0.000) | 0.016 (0.051) | **0.002 (0.005)** | 0.003 (0.005) | 0.000 (0.000) | 0.004 (0.014) | **0.002 (0.004)** | **0.002 (0.004)** |
| 1000 | AF | **0.010 (0.012)** | 0.010 (0.012) | 0.037 (0.109) | 0.013 (0.020) | **0.008 (0.013)** | 0.008 (0.013) | 0.038 (0.118) | 0.052 (0.163) | **0.008 (0.014)** | **0.007 (0.013)** |
| | SF | **0.003 (0.007)** | 0.003 (0.007) | 0.001 (0.001) | 0.017 (0.052) | **0.002 (0.005)** | 0.002 (0.005) | 0.001 (0.003) | 0.005 (0.016) | **0.002 (0.004)** | **0.001 (0.003)** |
| | Pro. | **100.0** | 100.0 | 100.0 | 97.5 | **100.0** | 100.0 | 100.0 | 0.4 | **100.0** | **100.0** |
| | Time | **0.006** | 13.797 | 21.965 | 142.390 | **0.037** | 2957.455 | 3086.742 | 8583.513 | **0.043** | **0.119** |

Table 5: Comparison of the BCME estimators ($\hat{\mathbf{R}}$, $\hat{\mathbf{\Sigma}}$), TP estimators ($\hat{\mathbf{R}}_{TP}$, $\hat{\mathbf{\Sigma}}_{TP}$), TP estimators with variable ordering ($\hat{\mathbf{R}}_{TP}^o$, $\hat{\mathbf{\Sigma}}_{TP}^o$), and QMLE estimators ($\hat{\mathbf{R}}_{QMLE}$, $\hat{\mathbf{\Sigma}}_{QMLE}$) of the blockwise correlation matrix and corresponding covariance matrix when the elements of $\epsilon_i$ follow a standardized exponential distribution. AS and AF represent the averages of the spectral-error and Frobenius-error, respectively. SS and SF denote the standard deviations of the spectral-error and Frobenius-error, respectively. Pro. (%) is the proportion of positive semi-definiteness. Time (in seconds) is the average execution time.

| $(K, p)$ | | (2,150) | | | | (4,420) | | | | (6,570) | (8,840) |
|---|---|---|---|---|---|---|---|---|---|---|---|
| $n$ | Measures | $\hat{\mathbf{\Sigma}}(\hat{\mathbf{R}})$ | $\hat{\mathbf{\Sigma}}_{TP}^o(\hat{\mathbf{R}}_{TP}^o)$ | $\hat{\mathbf{\Sigma}}_{TP}(\hat{\mathbf{R}}_{TP})$ | $\hat{\mathbf{\Sigma}}_{QMLE}(\hat{\mathbf{R}}_{QMLE})$ | $\hat{\mathbf{\Sigma}}(\hat{\mathbf{R}})$ | $\hat{\mathbf{\Sigma}}_{TP}^o(\hat{\mathbf{R}}_{TP}^o)$ | $\hat{\mathbf{\Sigma}}_{TP}(\hat{\mathbf{R}}_{TP})$ | $\hat{\mathbf{\Sigma}}_{QMLE}(\hat{\mathbf{R}}_{QMLE})$ | $\hat{\mathbf{\Sigma}}(\hat{\mathbf{R}})$ | $\hat{\mathbf{\Sigma}}(\hat{\mathbf{R}})$ |
| | AS | **0.021 (0.024)** | 0.021 (0.024) | 0.036 (0.108) | 0.029 (0.054) | **0.016 (0.025)** | 0.016 (0.025) | 0.027 (0.086) | 0.041 (0.125) | **0.015 (0.025)** | **0.011 (0.022)** |
| | SS | **0.009 (0.014)** | 0.009 (0.014) | 0.002 (0.000) | 0.028 (0.091) | **0.006 (0.011)** | 0.006 (0.011) | 0.002 (0.000) | 0.015 (0.048) | **0.005 (0.010)** | **0.003 (0.007)** |
| 200 | AF | **0.023 (0.026)** | 0.023 (0.026) | 0.042 (0.112) | 0.031 (0.057) | **0.018 (0.030)** | 0.018 (0.030) | 0.041 (0.120) | 0.053 (0.159) | **0.018 (0.031)** | **0.014 (0.029)** |
| | SF | **0.008 (0.015)** | 0.008 (0.015) | 0.005 (0.004) | 0.029 (0.093) | **0.005 (0.012)** | 0.005 (0.012) | 0.002 (0.005) | 0.016 (0.055) | **0.004 (0.009)** | **0.003 (0.007)** |
| | Pro. | **100.0** | 100.0 | 100.0 | 93.7 | **100.0** | 100.0 | 100.0 | 9.4 | **100.0** | **100.0** |
| | Time | **0.003** | 12.870 | 20.194 | 31.298 | **0.024** | 2717.192 | 2764.434 | 1631.160 | **0.031** | **0.091** |
| | AS | **0.013 (0.015)** | 0.013 (0.015) | 0.035 (0.108) | 0.018 (0.031) | **0.010 (0.016)** | 0.010 (0.017) | 0.026 (0.086) | 0.040 (0.126) | **0.010 (0.016)** | **0.007 (0.014)** |
| | SS | **0.005 (0.009)** | 0.005 (0.009) | 0.000 (0.000) | 0.022 (0.070) | **0.004 (0.008)** | 0.004 (0.008) | 0.001 (0.000) | 0.009 (0.029) | **0.003 (0.007)** | **0.002 (0.005)** |
| 500 | AF | **0.014 (0.016)** | 0.014 (0.016) | 0.038 (0.110) | 0.019 (0.033) | **0.012 (0.019)** | 0.012 (0.019) | 0.039 (0.118) | 0.053 (0.164) | **0.011 (0.020)** | **0.009 (0.019)** |
| | SF | **0.005 (0.009)** | 0.005 (0.009) | 0.002 (0.002) | 0.023 (0.072) | **0.003 (0.007)** | 0.003 (0.007) | 0.001 (0.003) | 0.010 (0.031) | **0.003 (0.006)** | **0.002 (0.004)** |
| | Pro. | **100.0** | 100.0 | 100.0 | 95.6 | **100.0** | 100.0 | 100.0 | 2.0 | **100.0** | **100.0** |
| | Time | **0.004** | 13.153 | 21.025 | 74.586 | **0.028** | 2787.388 | 2875.737 | 4011.246 | **0.035** | **0.100** |
| | AS | **0.009 (0.011)** | 0.009 (0.011) | 0.035 (0.108) | 0.013 (0.021) | **0.007 (0.011)** | 0.007 (0.012) | 0.026 (0.086) | 0.040 (0.125) | **0.007 (0.011)** | **0.005 (0.010)** |
| | SS | **0.004 (0.007)** | 0.004 (0.007) | 0.002 (0.007) | 0.018 (0.055) | **0.002 (0.005)** | 0.002 (0.005) | 0.000 (0.000) | 0.007 (0.023) | **0.002 (0.004)** | **0.002 (0.003)** |
| 1000 | AF | **0.010 (0.012)** | 0.010 (0.012) | 0.037 (0.109) | 0.014 (0.022) | **0.008 (0.013)** | 0.008 (0.014) | 0.038 (0.118) | 0.052 (0.163) | **0.008 (0.014)** | **0.006 (0.013)** |
| | SF | **0.004 (0.007)** | 0.004 (0.007) | 0.003 (0.008) | 0.018 (0.056) | **0.002 (0.005)** | 0.002 (0.005) | 0.001 (0.003) | 0.008 (0.025) | 0.002 (0.004) | **0.001 (0.003)** |
| | Pro. | **100.0** | 100.0 | 100.0 | 97.4 | **100.0** | 100.0 | 100.0 | 0.6 | **100.0** | **100.0** |
| | Time | **0.006** | 13.638 | 21.743 | 142.581 | **0.037** | 2984.105 | 3084.462 | 8654.350 | **0.043** | **0.118** |

Table 6: Results of block number selection when the elements of $\epsilon_i$ follow a mixture normal distribution $0.9\mathcal{N}(0, 5/9) + 0.1\mathcal{N}(0, 5)$. CT is the average percentage of the correct fit. Mean is the mean of the estimated number of blocks.

|  | $n = 200$ | | $n = 500$ | | $n = 1000$ | |
|---|---|---|---|---|---|---|
|  | CT | Mean | CT | Mean | CT | Mean |
| $K = 2$ | 1.00 | 2.00 | 1.00 | 2.00 | 1.00 | 2.00 |
| $K = 3$ | 1.00 | 3.00 | 1.00 | 3.00 | 1.00 | 3.00 |
| $K = 4$ | 0.93 | 3.78 | 1.00 | 4.00 | 1.00 | 4.00 |
| $K = 5$ | 0.59 | 3.36 | 1.00 | 5.00 | 1.00 | 5.00 |
| $K = 6$ | 0.13 | 1.67 | 0.99 | 5.94 | 1.00 | 6.00 |
| $K = 7$ | 0.03 | 1.19 | 0.89 | 6.36 | 1.00 | 7.00 |
| $K = 8$ | 0.03 | 1.22 | 0.94 | 7.55 | 1.00 | 8.00 |

Table 7: Results of block number selection when the elements of $\epsilon_i$ follow a standardized exponential distribution. CT is the average percentage of the correct fit. Mean is the mean of the estimated number of blocks.

|  | $n = 200$ | | $n = 500$ | | $n = 1000$ | |
|---|---|---|---|---|---|---|
|  | CT | Mean | CT | Mean | CT | Mean |
| $K = 2$ | 1.00 | 2.00 | 1.00 | 2.00 | 1.00 | 2.00 |
| $K = 3$ | 1.00 | 3.00 | 1.00 | 3.00 | 1.00 | 3.00 |
| $K = 4$ | 0.90 | 3.71 | 1.00 | 4.00 | 1.00 | 4.00 |
| $K = 5$ | 0.58 | 3.33 | 1.00 | 5.00 | 1.00 | 5.00 |
| $K = 6$ | 0.13 | 1.64 | 0.99 | 5.94 | 1.00 | 6.00 |
| $K = 7$ | 0.03 | 1.16 | 0.90 | 6.41 | 1.00 | 7.00 |
| $K = 8$ | 0.03 | 1.22 | 0.93 | 7.52 | 1.00 | 8.00 |

Table 8: The performance of the BCME estimators ($\hat{R}_{\hat{\Theta}}$, $\hat{\Sigma}_{\hat{\Theta}}$) of the blockwise correlation matrix and corresponding covariance matrix with given $K$ when the group memberships are unknown and $\epsilon_i$ follows a multivariate normal distribution $\mathcal{N}_p(\mathbf{0}_p, \mathbf{I}_p)$. AS and AF represent the averages of the spectral-error and Frobenius-error, respectively. SS and SF denote the standard deviations of the spectral-error and Frobenius-error, respectively. Pro. (%) is the proportion of positive semi-definiteness. Time (in seconds) is the average execution time.

| $(K, p)$ | | (2,150) | | (4,420) | | (6,570) | | (8,840) | |
|---|---|---|---|---|---|---|---|---|---|
| $n$ | Measures | $\Sigma(R)$ | $\Sigma_{\hat{\Theta}}(R_{\hat{\Theta}})$ | $\Sigma(R)$ | $\Sigma_{\hat{\Theta}}(R_{\hat{\Theta}})$ | $\Sigma(R)$ | $\Sigma_{\hat{\Theta}}(R_{\hat{\Theta}})$ | $\Sigma(R)$ | $\Sigma_{\hat{\Theta}}(R_{\hat{\Theta}})$ |
| 200 | AS | 0.019 (0.023) | 0.019 (0.027) | 0.014 (0.025) | 0.017 (0.031) | 0.014 (0.025) | 0.016 (0.031) | 0.010 (0.022) | 0.012 (0.030) |
|  | SS | 0.009 (0.013) | 0.009 (0.017) | 0.006 (0.012) | 0.007 (0.018) | 0.006 (0.010) | 0.006 (0.015) | 0.004 (0.007) | 0.004 (0.012) |
|  | AF | 0.020 (0.025) | 0.021 (0.031) | 0.016 (0.030) | 0.019 (0.036) | 0.016 (0.031) | 0.019 (0.037) | 0.013 (0.029) | 0.015 (0.038) |
|  | SF | 0.008 (0.014) | 0.009 (0.018) | 0.005 (0.012) | 0.007 (0.019) | 0.005 (0.010) | 0.006 (0.015) | 0.003 (0.007) | 0.004 (0.013) |
|  | Pro. | 100 | 100 | 100 | 100 | 100 | 100 | 100 | 100 |
|  | Time | 0.004 | 0.002 | 0.023 | 0.008 | 0.031 | 0.016 | 0.091 | 0.046 |
| 500 | AS | 0.012 (0.014) | 0.013 (0.017) | 0.009 (0.016) | 0.011 (0.021) | 0.009 (0.016) | 0.009 (0.021) | 0.006 (0.014) | 0.008 (0.021) |
|  | SS | 0.006 (0.009) | 0.006 (0.010) | 0.004 (0.007) | 0.006 (0.018) | 0.003 (0.006) | 0.004 (0.014) | 0.002 (0.005) | 0.004 (0.012) |
|  | AF | 0.013 (0.016) | 0.015 (0.020) | 0.010 (0.019) | 0.012 (0.024) | 0.010 (0.020) | 0.011 (0.025) | 0.008 (0.019) | 0.010 (0.025) |
|  | SF | 0.005 (0.009) | 0.006 (0.012) | 0.003 (0.007) | 0.006 (0.018) | 0.003 (0.006) | 0.004 (0.014) | 0.002 (0.004) | 0.004 (0.012) |
|  | Pro. | 100 | 100 | 100 | 100 | 100 | 100 | 100 | 100 |
|  | Time | 0.004 | 0.002 | 0.028 | 0.009 | 0.034 | 0.018 | 0.100 | 0.050 |
| 1000 | AS | 0.008 (0.010) | 0.009 (0.011) | 0.006 (0.011) | 0.008 (0.018) | 0.006 (0.011) | 0.007 (0.014) | 0.005 (0.010) | 0.006 (0.014) |
|  | SS | 0.004 (0.006) | 0.004 (0.007) | 0.003 (0.005) | 0.006 (0.019) | 0.002 (0.004) | 0.004 (0.012) | 0.002 (0.003) | 0.003 (0.011) |
|  | AF | 0.009 (0.011) | 0.009 (0.013) | 0.007 (0.013) | 0.009 (0.020) | 0.007 (0.014) | 0.008 (0.017) | 0.006 (0.013) | 0.007 (0.018) |
|  | SF | 0.004 (0.007) | 0.004 (0.008) | 0.002 (0.005) | 0.006 (0.019) | 0.002 (0.004) | 0.004 (0.012) | 0.001 (0.003) | 0.003 (0.011) |
|  | Pro. | 100 | 100 | 100 | 100 | 100 | 100 | 100 | 100 |
|  | Time | 0.006 | 0.002 | 0.037 | 0.012 | 0.042 | 0.022 | 0.117 | 0.057 |

