**Response to Reviewer VQRz's official reviews on submission 10036**

We are grateful for your inspirational reviews and constructive suggestions. We have carefully revised the manuscript. In what follows, your reviews are shown in *italics*, which are then followed by our *point-by-point* responses.

Main Comments

(1) *The literature review on existing methods appears limited, which may lead to the proposed approach overlapping with established techniques. From the simple derivation below, the proposed blockwise correlation matrix estimation (BCME) in Equation (3) is closely related to Equation (12) in Engle and Kelly (2012). (Hat on $\hat{Y}$ zis omitted due to compilation issue.)*

$$\frac{\frac{1}{n}\sum_{i=1}^{n} e_{p_{k_1}}^{\top} Y_i^{(S_{k_1})} Y_i^{(S_{k_2})\top} e_{p_{k_2}}}{p_{k_1} p_{k_2}} = \frac{e_{p_{k_1}}^{\top} \left(\frac{1}{n}\sum_{i=1}^{n} Y_i^{(S_{k_1})} Y_i^{(S_{k_2})\top}\right) e_{p_{k_2}}}{p_{k_1} p_{k_2}}$$
$$= \frac{e_{p_{k_1}}^{\top} Cov(Y^{(S_{k_1},S_{k_2})}) e_{p_{k_2}}}{p_{k_1} p_{k_2}} = \frac{\sum_{i,j=1}^{N} \rho_{ij}}{p_{k_1} p_{k_2}}$$

*Furthermore, the issue of positive definiteness mentioned in the Introduction is examined in Corollary 2 of Archakov and Hansen (2024).*

REPLY: We appreciate your valuable reviews. According to your advice, we have cited the two references, Engle and Kelly (2012) and Archakov and Hansen (2024), and carefully compared the differences between their estimation methods with our BCME method to clarify the contribution of our paper. Specifically, compared with the Engle and Kelly (2012, EK), we have shown the following two differences.

(1a) **The different model.** Engle and Kelly (2012) imposed a blockwise structure on the Dynamic Conditional Correlation (DCC) models and proposed the Block DECO model. Specifically, the Block DECO correlations are calculated as the average DCC correlation within each block (i.e., EK's equation (12)). In other words, in addition to the blockwise structure, the Block DECO model incorporates additional prior structures. However, we focus solely on the blockwise correlation matrix without imposing any other structures, making our approach more general than EK's model.

(1b) **The different estimation methods and theoretical assumptions.** With the assumptions that the variables follow a Gaussian distribution and the dimension of

variables is fixed, Engle and Kelly (2012) employed the maximum likelihood estimation for the Block DECO model and showed the asymptotic properties of the estimated parameters. In contrast, we develop a novel closed-form estimator for the blockwise correlation matrix of variables using moment method (see, equation (3) in our paper). And we establish the asymptotic normality of the estimated parameters under certain moment conditions and allowing the dimension of the variables to exceed the sample size.

In short, the equation (3) in our paper is the estimator for the blockwise correlation matrix and the EK's equation (12) is a DCC model with a blockwise correlation matrix, which are not closely related.

Next, after carefully studying the article of Archakov and Hansen (2024, AH), we have found that Archakov and Hansen (2024) did examine the problem of positive definiteness mentioned in our Introduction by their Corollary 2. Then, the positive semi-definiteness of the estimated blockwise correlation matrix can simply be verified by their Corollary 2. In our paper, we similarly obey the AH's Corollary 2 and ensure the positive semi-definiteness of the estimated blockwise correlation matrix and their corresponding covariance matrix. In addition to this point, we have shown the following one difference between our paper and theirs.

(1c) **The different goals.** The main goals of Archakov and Hansen (2024) are to derive a canonical representation for a broad class of block matrices which includes the blockwise correlation matrices as the special cases. This canonical representation simplifies the computations of several matrix function, which improves the maximum likelihood estimation of a correlation matrix with the blockwise structure. However, our main goals are to correctly estimate the blockwise correlation matrix using moment method and establish its asymptotic properties.

In addition to the differences mentioned above, we have made the following one additional contribution.

(1d) **The block number determination and group membership recovery.** Engle and Kelly (2012) and Archakov and Hansen (2024) assumed that the block number and their memberships are given a priori. However, in real-world applications, their true values are unknown and need to be estimated correctly. We address this limitation by utilizing the ridge-type ratio criterion and spectral clustering to estimate the number of blocks and recover their memberships for a blockwise correlation matrix, and prove their consistency. Subsequently, we extend the asymptotic normality of

parameter estimators and stochastic convergence rate of the estimated blockwise correlation matrix and corresponding estimated covariance matrix to the scenario where the group memberships are unknown and the block number is given.

In sum, the three differences and one additional contribution mentioned above indicate our paper has considerable value; see lines 54-60 and 71-76 on page 2.

(2) *The notation is not rigorous. $Y_i$ inconsistently denotes the random variable and realization (sample) of the random variable.*

REPLY: Thank you for your careful reading and suggestion. After carefully studying the ICLR 2025 template, we have used more rigorous and consistent notation. Specifically, we have used $\mathbf{y}_i$ to represent the random variable and used $\boldsymbol{y}_i$ as the $i$-th realization of the random variable; see line 101 on page 2 and lines 273-274 on page 6, and also see our reply to the comment # 1 of Reviewer KthD and comment # 2 of Reviewer 4aHF.

(3) *The technical conditions lack intuitive explanation, making it unclear how practical they are in real-world applications.*

REPLY: We apologize for causing confusion, and have made our conditions more intuitive. Per your advice, we illustrate that the Condition (C1)(i) introduces the moment conditions of $\boldsymbol{\epsilon}_i$. And we imply that Condition (C1)(ii) ensuring that the distribution does not have "heavy tails" (e.g., Cauchy distribution). Furthermore, for condition (C2), we point out that it eliminates possible multicollinearity issues; see lines 171-181 on page 4.

(4) *The experiments omit the turnover ratio, a significant metric in portfolio optimization analysis.*

REPLY: Many thanks for your valuable review. We have added the turnover ratio in our paper; see Table 3 in the revised paper on page 8. Although the turnover ratio of the portfolio return based on BCME with RR is higher than that of the other portfolio return, the other measures have better performance, especially, Sharpe ratio. In short, the block structure is significant for portfolio management and our proposed framework is highly effective for portfolio analysis; see lines 367-377 on page 7 and lines 378-386 on page 8.

(5) *The topic somewhat diverges from the primary area of probabilistic methods.*

REPLY: Many thanks for your constructive reviews. The covariance matrix plays a fundamental role in probabilistic methods, particularly in statistical inference, parameter estimation, and model analysis. It is essential for describing the relationships, correlations, and variability between different random variables, making it a core component of many probabilistic approaches. Specifically, the covariance matrix has gained significant

popularity in various fields, including but not limited to: Machine learning (Bilmes 2000; Zhang and Rao 2013), finance and risk management (Markowitz 1952; Jagannathan and Ma 2003), econometrics (Chen and Conley 2001; Fan et al. 2008), biostatistics (Tong and Wang 2007; Friedman et al. 2008), and neuroscience and gene expression (Parketal. 2007; Wu and Smyth 2012; Tan et al. 2015; Eisenach et al. 2020; Pircalabelu and Claeskens 2020).

# Response to Reviewer Fohj's official reviews on submission 10036

We are grateful for your inspirational reviews and constructive suggestions. We have carefully revised the manuscript. In what follows, your reviews are shown in *italics*, which are then followed by our *point-by-point* responses.

## Main Comments

(1) *This paper's idea is essentially simultaneous clustering and estimation of a covariance matrix. Such an idea has been studied in some recent relevant literature; see, e.g., Su et al. (2016), Liu et al. (2020), Zhu et al. (2023), Liu et al. (2024). I think it will be helpful to discuss this line of research for a more complete picture.*

REPLY: Many thanks for your review. Actually, our paper's idea is a two-step estimation. First, we utilize the spectral clustering to recover the group membership $\mathbb{S}_k$s for $k = 1, \cdots, K$ when the number of blocks is predetermined. Second, we estimate the unknown parameters in the blockwise correlation matrix with the estimated group membership. This is different the recent relevant literature that simultaneously estimate the model parameters and group memberships with given $K$. Those method is theoretically complex and lacking generality, since their optimization functions are non-convex and require specific algorithms. We have highlighted the difference in the revised paper; see remark 1 on page 5.

(2) *The main theorem, Theorem 1, assumes that the group memberships for all variables are fully known. Later in the paper, however, these memberships are estimated through spectral clustering, which is shown to be consistent. I'd like to highlight a potential issue: the consistency established in Lei and Rinaldo (2015) indicates only that the percentage of mis-clustered nodes converges to zero in probability. For Theorem 1's asymptotic normality to hold, the rate of convergence would need this mis-clustered percentage to decrease faster than $n^{-1/2}$, which appears unachievable under current assumptions. Similar challenges are noted in references Liu et al. (2020) and Zhu et al. (2023), but these works resolve the issue by establishing almost sure convergence for group membership estimation, which may also be necessary here. Please elaborate on this issue.*

REPLY: Many thanks for your valuable review. Per your advice, we have added the almost sure convergence for the group membership estimation proved by Su et al. (2019), that is, for sufficiently large $n$ and $p$, $\sup_{1\leq i\leq n}\sup_{1\leq j\leq p}\mathbf{1}_{\{\hat{\boldsymbol{\theta}}_j\neq\boldsymbol{\theta}_j\}} = 0, a.s.$; see Condition (C5) on page 5. Based on this, we have obtained $\hat{\boldsymbol{\rho}}_{\hat{\boldsymbol{\Theta}}} \xrightarrow{p} \hat{\boldsymbol{\rho}}$, where $\hat{\boldsymbol{\rho}}_{\hat{\boldsymbol{\Theta}}}$ and $\hat{\boldsymbol{\rho}}$ are two estimators of

$\boldsymbol{\rho}$ with estimated and given group memberships, respectively. Then, we have extended the Theorem 1 to the case where group memberships are unknown; see Corollary 1 on page 5.

(3) *What happens if the estimated $\hat{K}$ is greater than the true $K$? Intuitively, it should still be ok as long as $\hat{K}$ is finite. For example, in Liu et al. (2020) and Zhu et al. (2023), the group estimators are still consistent even if $K$ is over-specified. Can you at least provide some simulation studies to investigate this issue? This would certainly add to the applicability of the proposed methodology.*

REPLY: Many thanks for your valuable review. According to your comment, we have conducted simulation studies with $\hat{K} = K + 1$ when the group memberships are unknown and $\boldsymbol{\epsilon}_i$ follows a multivariate normal distribution $\mathcal{N}_p(\mathbf{0}_p, \boldsymbol{I}_p)$. We obtain the same patterns as given $\hat{K} = K$, which implies that the blockwise correlation matrix and covariance matrix estimations are still consistent even if $K$ is over-specified; see Table R given on page 7 of the responses. In addition, the similar results are yielded when $\boldsymbol{\epsilon}_i$ follows non-normal distributions, but they are not reported here to save space.

Table R: The performance of the BCME estimators $(\hat{\boldsymbol{R}}_{\hat{\boldsymbol{\Theta}}}, \hat{\boldsymbol{\Sigma}}_{\hat{\boldsymbol{\Theta}}})$ of the blockwise correlation matrix and corresponding covariance matrix with $\hat{K} = K + 1$ when the group memberships are unknown and $\boldsymbol{\epsilon}_i$ follows a multivariate normal distribution $\mathcal{N}_p(\mathbf{0}_p, \boldsymbol{I}_p)$. AS and AF represent the averages of the spectral-error and Frobenius-error, respectively. SS and SF denote the standard deviations of the spectral-error and Frobenius-error, respectively. Pro. (%) is the proportion of positive semi-definiteness. Time (in seconds) is the average execution time.

| $(K,p)$ | | (2,150) | (4,420) | (6,570) | (8,840) |
|---|---|---|---|---|---|
| $n$ | Measures | $\hat{\boldsymbol{\Sigma}}_{\hat{\boldsymbol{\Theta}}}$ ($\hat{\boldsymbol{R}}_{\hat{\boldsymbol{\Theta}}}$) | $\hat{\boldsymbol{\Sigma}}_{\hat{\boldsymbol{\Theta}}}$ ($\hat{\boldsymbol{R}}_{\hat{\boldsymbol{\Theta}}}$) | $\hat{\boldsymbol{\Sigma}}_{\hat{\boldsymbol{\Theta}}}$ ($\hat{\boldsymbol{R}}_{\hat{\boldsymbol{\Theta}}}$) | $\hat{\boldsymbol{\Sigma}}_{\hat{\boldsymbol{\Theta}}}$ ($\hat{\boldsymbol{R}}_{\hat{\boldsymbol{\Theta}}}$) |
| | AS | 0.020 (0.033) | 0.016 (0.028) | 0.014 (0.028) | 0.011 (0.028) |
| | SS | 0.009 (0.016) | 0.006 (0.012) | 0.005 (0.014) | 0.004 (0.011) |
| | AF | 0.023 (0.042) | 0.018 (0.033) | 0.017 (0.035) | 0.014 (0.035) |
| 200 | SF | 0.009 (0.019) | 0.006 (0.013) | 0.005 (0.014) | 0.004 (0.011) |
| | Pro. | 100 | 100 | 100 | 100 |
| | Time | 0.001 | 0.008 | 0.028 | 0.045 |
| | AS | 0.014 (0.020) | 0.010 (0.018) | 0.009 (0.021) | 0.008 (0.020) |
| | SS | 0.006 (0.011) | 0.004 (0.011) | 0.004 (0.014) | 0.003 (0.011) |
| | AF | 0.016 (0.025) | 0.011 (0.022) | 0.011 (0.025) | 0.010 (0.025) |
| 500 | SF | 0.006 (0.014) | 0.004 (0.012) | 0.004 (0.014) | 0.004 (0.012) |
| | Pro. | 100 | 100 | 100 | 100 |
| | Time | 0.002 | 0.009 | 0.018 | 0.050 |
| | AS | 0.009 (0.013) | 0.007 (0.014) | 0.007 (0.016) | 0.007 (0.018) |
| | SS | 0.004 (0.008) | 0.003 (0.011) | 0.004 (0.014) | 0.004 (0.014) |
| | AF | 0.010 (0.015) | 0.008 (0.017) | 0.008 (0.019) | 0.008 (0.021) |
| 1000 | SF | 0.004 (0.011) | 0.004 (0.011) | 0.004 (0.014) | 0.004 (0.011) |
| | Pro. | 100 | 100 | 100 | 100 |
| | Time | 0.002 | 0.012 | 0.108 | 0.057 |

**Response to Reviewer KthD's official reviews on submission 10036**

We are grateful for your inspirational reviews and constructive suggestions. We have carefully revised the manuscript. In what follows, your reviews are shown in *italics*, which are then followed by our *point-by-point* responses.

Main Comments

(1) *The paper is challenging to read due to its dense presentation. The authors should make a substantial effort to improve clarity by introducing a dedicated notation section. This section should simplify the notations, provide clear definitions of each variable, and organize the symbols systematically for easy reference. Additionally, a more logical reorganization of the paper's sections would enhance readability.*

REPLY: We apologize for causing confusion, and have made our statements clearer. Per your advice, after carefully studying the ICLR 2025 template, we first introduce a dedicated notation section 2.1 to simplify and clarify the notation throughout the paper. For example, vectors are denoted by lower-case bold letters, e.g., $\boldsymbol{\iota} = (\iota_1, \cdots, \iota_m)^\top \in \mathbb{R}^m$, and matrices by upper-case bold, e.g., $\boldsymbol{M} = (M_{ij}) \in \mathbb{R}^{m \times m}$; see lines 90-100 on page 2, and also see our reply to the comment # 2 of Reviewer VQRz and Reviewer 4aHF.

Next, we have reorganized our paper's original section 2 to reduce the dense presentation. Specifically, we have set original section 2.1 as the new section 2 and subdivided the new section 2 into section 2.1 basic notations and definition, section 2.2 blockwise correlation matrix estimation, and section 2.3 asymptotic analysis. And the original section 2.2 have been set as the new section 3; see lines 77-84 on page 2.

(2) *The experimental results are also difficult to interpret. The tables do not clearly indicate which method performs best; highlighting the best values in bold would improve clarity. Furthermore, the presentation of results would benefit from including statistical tests, such as p-values, to provide a more robust comparison between methods. Also the other should include more baselines as comparisons since a lot of covariance estimators have recently been developed in Random Matrix Theory and statistical physics ( linear and non linear shrinkage of ledoit Wolf,...)*

REPLY: We apologize for causing confusion again, and have made our experimental results clearer. Per your advice, in the simulation and real data, we have highlighted the results of our method in bold. Moreover, in real data, we have also highlighted the best values for

each measure to indicate which method performs best for that measure. Then, the block structure is significant for portfolio management and our proposed framework is highly effective for portfolio analysis. see Table 1 on page 6, Table 3 on page 8, and Tables 4-5 on page 17.

In addition, we have added the $p$-value for Beta in Table 3 on page 8, which shows that all method are significant. This, together with the lowest value of Beta for our method, implies that our method exhibits lower risk than other methods; see line 377 on page 7 and line 378 on page 8.

Finally, we have included three additional methods to compared with our method, that is, the methods of Ledoit and Wolf (2003), Ledoit and Wolf (2020), and Schäfer and Strimmer (2005). The results in Table 3 on page 8 indicate the additional three methods in each measure are lower than our method. Then, we can clearly find that our method significantly outperforms other methods; see lines 352-367 and 375-377 on page 7, lines 378-386 on page 8, and also see our reply to the comment # 3 of Reviver XHvq.

**Response to Reviewer XHvq's official reviews on submission 10036**

We are grateful for your inspirational reviews and constructive suggestions. We have carefully revised the manuscript. In what follows, your reviews are shown in *italics*, which are then followed by our *point-by-point* responses.

Main Comments

(1) *While the empirical study shows that the proposed method outperforms existing methods in portfolio optimization, the paper lacks a clear explanation or theoretical justification for why this is the case. It remains unclear how the statistical properties of the proposed estimator translate into better portfolio performance.*

REPLY: Many thanks for your valuable review. In Ledoit and Wolf (2004, LW), the covariance matrix estimator is $\hat{\boldsymbol{\Sigma}}_{LW} = \hat{\gamma}\hat{\mu}\boldsymbol{I}_p + (1 - \hat{\gamma})\boldsymbol{S}$, where $\hat{\gamma} \in [0, 1]$, $\hat{\mu} = tr(\boldsymbol{S})/p$, and $\boldsymbol{S}$ is the sample covariance matrix. It is worth noting that $\boldsymbol{S}$ is singular matrix for $p > n$ and $\hat{\boldsymbol{\Sigma}}_{LW}$ approaches $\hat{\gamma}\hat{\mu}\boldsymbol{I}_p$ when $p$ increases faster than $n$. Hence, LW's covariance matrix estimator can be considered a special case ($K = 1$) of our estimator when $p >> n$, which reveals our method outperforms LW's method. In Tsay and Pourahmadi (2017, TP), although the block structure reduces the estimated number of angle parameters (i.e., From the angle matrix $O(p^2)$ to pivotal angles $O(K^2)$) via MLE, restoring the estimated correlation matrix still requires that the angle matrix, which is a tough task when $p$ diverges. However, our estimator is closed-form, which is significantly faster than TP's method.

(2) *The paper could offer more intuitive explanations or theoretical insights into why the proposed estimator is expected to perform better in applications like portfolio optimization. Connecting the methodological advancements to practical outcomes would enhance the paper's impact.*

REPLY: Many thanks for your valuable review. We have listed two advantages of our estimator in portfolio optimization. First, the low rank assumption that correlation matrix has a blockwise structure effectively reduces the number of unknown parameters in the covariance matrix model from $O(p^2)$ to $O(p + K^2)$. This will yield a covariance matrix estimator with smaller errors and improve the robustness of the optimal weights in the high-dimensional portfolio optimization. Second, Blondes et al. (2013) and Tsay and Pourahmadi (2017) indicated that the sample correlation matrix of stock returns exhibits

structured patterns. Then, for the block number estimation and their membership recovery, our data-driven method better captures the block structure compared to directly specifying it; see Table 3 on page 8.

(3) *The comparisons in the empirical study are primarily with the TP method and the Ledoit-Wolf estimator. Including a wider range of contemporary high-dimensional covariance estimation methods in the comparison would provide a more comprehensive evaluation of the proposed method's performance.*

REPLY: Many thanks for your valuable review. Per your advice, we have considered three additional methods to estimate the high-dimensional covariance matrix, that is, the methods of Ledoit and Wolf (2003), Ledoit and Wolf (2020), and Schäfer and Strimmer (2005). The results in Table 3 on page 8 indicate the additional three methods in each measure are lower than our method. Then, we can clearly find that our method significantly outperforms other methods; see lines 352-367 and 375-377 on page 7, lines 378-386 on page 8, and also see our reply to the comment # 2 of Reviver KthD.

# Response to Reviewer 4aHF's official reviews on submission 10036

We are grateful for your inspirational reviews and constructive suggestions. We have carefully revised the manuscript. In what follows, your reviews are shown in *italics*, which are then followed by our *point-by-point* responses.

## Main Comments

(1) *In the abstract, the authors state that "without imposing any distribution assuptions". This statement lacks precision, as certain technical conditions are required.*

REPLY: Thank you for your careful reading and suggestion. Per your advice, we have replaced the previous expression "without imposing any distribution assumptions" with "under certain moment conditions " throughout the paper; see line 16 on page 1, line 69 on page 2, and line 415 on page 8.

(2) *The notation is somewhat heavy, please consider improving the presentation for clarity.*

REPLY: We appreciate your valuable review. Per your advice, we have simplified and clarified the notation throughout the paper with carefully studying the ICLR 2025 template. For example, vectors are denoted by lower-case bold letters, e.g., $\boldsymbol{\iota} = (\iota_1, \cdots, \iota_m)^\top \in \mathbb{R}^m$, and matrices by upper-case bold, e.g., $\boldsymbol{M} = (M_{ij}) \in \mathbb{R}^{m \times m}$; see section 2.1 on pages 2-3, and also see our reply to the comment # 2 of Reviewer VQRz and comment # 1 of Reviewer KthD.

(3) *In Theorem 1, how should the convergence result be interpreted as the dimension $p$ approaches infinity? Discuss the possible connection with high-dimensional Gaussian approximation.*

REPLY: Many thanks for your review. In equation (3), we can find that $\hat{\rho}_{k_1 k_2}$ for $k_1, k_2 = 1, \cdots, K$ depends on $\hat{\sigma}_j^2$ for $j = 1, \cdots, p$. This, together with the Lemma 3 in Appendix A, implies that as long as $(\log p)^{6/\gamma_1 - 1} = o(n)$ holds, as $\min\{n, p\} \to \infty$, $\hat{\sigma}_j^2 \overset{p}{\to} \sigma_j^2$ and $\hat{\rho}_{k_1 k_2} \overset{p}{\to} \rho_{k_1 k_2}$. Although the condition $(\log p)^{6/\gamma_1 - 1} = o(n)$ allows $p >> n$, our model has no connection with high-dimensional Gaussian approximation. The reason is that we impose a low rank structure, the blockwise structure, on the correlation matrix. Then, our model essentially belongs to the domain of dimensionality reduction; see equation (3) on page 3 and Lemma 3 on page 12.

(4) *Please provide references for the comparative methods in the experimental studies. Moreover, why are the results of BCME and TP exactly identical?*

REPLY: Many thanks for your review, and we apologize for causing confusion. Per you advice, we have added the references for the comparative methods in the real data. In addition, we have replaced the previous expression "We also employ the TP method used in our simulation studies" with "we employ the Tsay and Pourahmadi (2017,TP)'s method with variable ordering". Since we have sorted the variables before using the TP method, we obtain the same result of BCME and TP methods. This find can also be discovered in Table 1 in the simulation; see line 355 on page 7.

(5) *What are the results under different values of $K$ with a fixed dimension $p$?*

REPLY: Many thanks for your review. We have presented the different values of $K$ with a fixed dimension $p$ in Section 5; see Table 3 on page 8. This implies that the results of using different $K$ are similar, but the result of $K$ obtained by data-driven is the best; see lines 375-377 on page 7 and lines 378-386 on page 8.

(6) *It would be helpful to include an arrow indicating whether a larger metric corresponds to better performance.*

REPLY: All done; see Table 3 on page 8.