# OpenReview forum: "Inferences on Covariance Matrix with Blockwise Correlation Structure"
_ICLR.cc/2025/Conference — Submitted to ICLR 2025_

### Official Review · Reviewer_4aHF · 2024-11-04

**Soundness:** 3
**Presentation:** 3
**Contribution:** 3
**Rating:** 6
**Confidence:** 4

**Summary:**

This paper proposes a closed-form estimator for the blockwise correlation matrix by utilizing the sample moments of variable means within groups. It also addresses the scenario where the block number and group memberships of the variables are unknown.

**Strengths:**

The paper is well organized. It demonstrates the performance of the proposed method through both theoretical and numerical analyses.

**Weaknesses:**

Please refer to the questions below.

**Questions:**

1. In the abstract, the authors state that "without imposing any distribution assuptions". This statement lacks precision, as certain technical conditions are required.

2. The notation is somewhat heavy, please consider improving the presentation for clarity.

3. In Theorem 1, how should the convergence result be interpreted as the dimension $p$ approaches infinity? Discuss the possible connection with high-dimensional Gaussian approximation.

4. Please provide references for the comparative methods in the experimental studies. Moreover, why are the results of BCME and TP exactly identical?

5. What are the results under different values of $K$ with a fixed dimension $p$?

6. It would be helpful to include an arrow indicating whether a larger metric corresponds to better performance.

---

> ### Author Response · Authors · 2024-11-27
> **Reply to Question 1**
>
> Thank you for your careful reading and suggestion.  Per your advice, we have
> 	replaced the previous expression ''without imposing any distribution assumptions'' with ''under certain moment conditions'' throughout the paper; see line 16 on page 1, line 69 on page 2, and line 415 on page 8.

---

> ### Author Response · Authors · 2024-11-27
> **Reply to Question 2**
>
> We appreciate your valuable review.
> 	Per your advice, we have simplified and clarified the notation throughout the paper with carefully studying the ICLR 2025 template. For example, vectors are denoted by lower-case bold letters, e.g., $\boldsymbol{\iota}=(\iota_1,\cdots,\iota_m)^\top\in\mathbb{R}^{m}$, and matrices by upper-case bold, e.g., $\symbfit{M}=(M_{ij})\in\mathbb{R}^{m\times m}$; see section 2.1 on pages 2-3, and also see our reply to the comment \# 2 of Reviewer VQRz and comment \# 1 of Reviewer KthD.

---

> ### Author Response · Authors · 2024-11-27
> **Reply to Question 3**
>
> Many thanks for your  review. In equation (3), we can find that $\hat\rho_{k_1k_2}$ for $k_1,k_2=1,\cdots,K$ depends on $\hat\sigma_j^2$ for $j=1,\cdots, p$. This, together with the Lemma 3 in Appendix A, implies that as long as $(\log p)^{6/\gamma_1-1}=o(n)$ holds, as $\min\{n,p\}\rightarrow\infty$, $\hat\sigma_j^2\stackrel{p}{\rightarrow}\sigma_j^2$ and $\hat\rho_{k_1k_2}\stackrel{p}{\rightarrow}\rho_{k_1k_2}$.
> 	Although the condition $(\log p)^{6/\gamma_1-1}=o(n)$ allows $p>>n$, our model has no connection with high-dimensional Gaussian approximation. The reason is that we impose a low rank structure, the blockwise structure, on the correlation matrix. Then, our model essentially belongs to the domain of dimensionality reduction; see equation (3) on page 3 and Lemma 3 on page 12.

---

> ### Author Response · Authors · 2024-11-27
> **Reply to Question 4**
>
> Many thanks for your  review, and we apologize for causing confusion.
> 	Per you advice, we have added the references for the comparative methods in the real data. In addition, we have
> 	replaced the previous expression ''We also employ the TP method
> 	used in our simulation studies'' with ''we employ the Tsay and Pourahmadi (2017,TP)’s method with variable ordering''.
> 	Since we have sorted the variables before using the TP method, we obtain the same result of BCME and TP methods. This find can also be discovered in Table 1 in the simulation;
> 	 see line 355 on page 7.

---

> ### Author Response · Authors · 2024-11-27
> **Reply to Question 5**
>
> Many thanks for your review. We have presented the different values of $K$ with a fixed dimension $p$ in Section 5; see Table 3 on page 8. This implies that the results of using different $K$ are similar, but the result of $K$ obtained by data-driven is the best; see lines 375-377 on page 7 and lines 378-386 on page 8.

---

> ### Author Response · Authors · 2024-11-27
> **Reply to Question 6**
>
> All done; see Table 3 on page 8.

---

### Official Review · Reviewer_XHvq · 2024-11-04

**Soundness:** 2
**Presentation:** 3
**Contribution:** 2
**Rating:** 5
**Confidence:** 3

**Summary:**

The paper presents a closed-form estimator for the blockwise correlation matrix of high-dimensional data without imposing specific distributional assumptions. By utilizing sample moments of variable means within known groups, the method ensures the positive semi-definiteness of the estimated covariance matrix without requiring predetermined variable ordering. The authors also introduce a ridge-type ratio criterion to estimate the number of blocks and employ spectral clustering to recover group memberships, proving the consistency of these approaches. Extensive simulations and an empirical study on Chinese stock market returns demonstrate the method's effectiveness.

**Strengths:**

1. The introduction of a closed-form estimator for the blockwise correlation matrix addresses the challenges posed by high-dimensional data, such as the singularity of the sample covariance matrix when $p > n$.

2. The paper provides thorough theoretical analysis, including proofs of asymptotic normality and convergence rates without relying on stringent distributional assumptions, enhancing the robustness and general applicability of the proposed method.

3. The method guarantees the positive semi-definiteness of the estimated covariance matrix without necessitating a predetermined variable order, overcoming limitations of some existing methods like the TP estimator.

**Weaknesses:**

1. While the empirical study shows that the proposed method outperforms existing methods in portfolio optimization, the paper lacks a clear explanation or theoretical justification for why this is the case. It remains unclear how the statistical properties of the proposed estimator translate into better portfolio performance.

2. The paper could offer more intuitive explanations or theoretical insights into why the proposed estimator is expected to perform better in applications like portfolio optimization. Connecting the methodological advancements to practical outcomes would enhance the paper's impact.

3. The comparisons in the empirical study are primarily with the TP method and the Ledoit-Wolf estimator. Including a wider range of contemporary high-dimensional covariance estimation methods in the comparison would provide a more comprehensive evaluation of the proposed method's performance.

**Questions:**

Please refer to the Weaknesses section.

---

> ### Author Response · Authors · 2024-11-27
> **Reply Weakness 1**
>
> Many thanks for your valuable review.
> In Ledoit and Wolf (2004, LW), the covariance matrix estimator is
> $$\hat{\boldsymbol{\Sigma}}_{LW}=\hat {\gamma}\hat{\mu}\symbfit{I}_p+(1-\hat{\gamma})\symbfit{S},$$
>
> where $\hat{\gamma}\in[0,1]$, $\hat{\mu}=tr(\symbfit{S})/p$, and $\symbfit{S}$ is the sample covariance matrix.
> It is worth noting that $\symbfit{S}$ is singular matrix for $p>n$ and $\hat{\boldsymbol{\Sigma}}_{LW}$ approaches $\hat \gamma\hat\mu\symbfit{I}_p$ when $p$ increases faster than $n$. Hence, LW's covariance matrix estimator can be considered a special case ($K=1$) of our estimator when $p>>n$, which reveals our method outperforms LW's method.
> In Tsay and Pourahmadi (2017, TP), although the block structure reduces the estimated number of angle parameters (i.e., From the angle matrix $O(p^2)$ to pivotal angles $O(K^2)$) via MLE, restoring the estimated correlation matrix still requires that the angle matrix, which is a tough task when $p$ diverges. However, our estimator is closed-form, which is significantly faster than TP's method.

---

> ### Author Response · Authors · 2024-11-27
> **Reply to Weakness 2**
>
> Many thanks for your valuable review. We have listed two advantages of our estimator in portfolio optimization.
> First, the low rank assumption that correlation matrix has a blockwise structure effectively reduces the number of unknown parameters in the covariance matrix model from $O(p^2)$ to $O(p+K^2)$. This will yield a covariance matrix estimator with smaller errors and improve the robustness of the optimal weights in the high-dimensional portfolio optimization.
> Second, Blondes et al. (2013) and Tsay and Pourahmadi (2017) indicated that the sample correlation matrix of stock returns exhibits structured patterns. Then, for the block number estimation and their membership recovery, our data-driven method better captures the block structure compared to directly specifying it; see Table 3 on page 8.

---

> ### Author Response · Authors · 2024-11-27
> **Reply to Weakness 3**
>
> Many thanks for your valuable review. Per your advice, we have considered  three additional methods to estimate the high-dimensional covariance matrix, that is, the methods of
> 	Ledoit and Wolf (2003), Ledoit and Wolf
> 	(2020), and Sch{\"a}fer and Strimmer (2005).
> 	The results in Table 3 on page 8 indicate the additional three methods in each measure are lower than our method. Then, we can clearly find that our method significantly outperforms other methods; see lines 352-367 and 375-377 on page 7, lines 378-386 on page 8, and also see our reply to the comment \# 2 of Reviver KthD.

---

### Official Review · Reviewer_KthD · 2024-11-05

**Soundness:** 2
**Presentation:** 1
**Contribution:** 2
**Rating:** 3
**Confidence:** 2

**Summary:**

The paper presents a closed-form estimator for the blockwise correlation matrix of variables that ensures positive semi-definiteness and is suitable for high-dimensional data. The authors demonstrate consistency and asymptotic properties without distributional assumptions and use spectral clustering to identify block structures. Their method is tested with simulations and applied to stock data from the Chinese market.

**Strengths:**

1- The problem solved by the authors are interesting
2- The method comes with theoretical guarantees

**Weaknesses:**

The paper is challenging to read due to its dense presentation. The authors should make a substantial effort to improve clarity by introducing a dedicated notation section. This section should simplify the notations, provide clear definitions of each variable, and organize the symbols systematically for easy reference. Additionally, a more logical reorganization of the paper’s sections would enhance readability.

The experimental results are also difficult to interpret. The tables do not clearly indicate which method performs best; highlighting the best values in bold would improve clarity. Furthermore, the presentation of results would benefit from including statistical tests, such as p-values, to provide a more robust comparison between methods. Also the other should include more baselines as comparisons since a lot of covariance estimators have recently been developed in Random Matrix Theory and statistical physics ( linear and non linear shrinkage of ledoit Wolf,...)

**Questions:**

1- How does this method compared to other covariance estimator ( Linear and non linear shrinkage of Ledoit Wolf,...)?

---

> ### Author Response · Authors · 2024-11-27
> **Reply to Weakness 1**
>
> We apologize for causing confusion, and have made our statements clearer. Per your advice, after carefully studying the ICLR 2025 template, we first introduce a dedicated notation section 2.1 to simplify and clarify the notation throughout the paper. For example, vectors are denoted by lower-case bold letters, e.g., $\boldsymbol{\iota}=(\iota_1,\cdots,\iota_m)^\top\in\mathbb{R}^{m}$, and matrices by upper-case bold, e.g., $\symbfit{M}=(M_{ij})\in\mathbb{R}^{m\times m}$; see lines 90-100 on page 2, and also see our reply to the comment \# 2 of Reviewer VQRz and Reviewer 4aHF.
>
> Next, we have reorganized our paper's original section 2 to reduce the dense presentation. Specifically, we have set original section 2.1 as the new section 2 and subdivided the new section 2 into section 2.1 basic notations and definition, section 2.2 blockwise correlation matrix estimation, and section 2.3 asymptotic analysis. And the original section 2.2 have been set as the new section 3; see lines 77-84 on page 2.

---

> ### Author Response · Authors · 2024-11-27
> **Reply to Weakness 2**
>
> We apologize for causing confusion again, and have made our experimental results clearer. Per your advice, in the simulation and real data, we have highlighted the results of our method in bold.
> Moreover, in real data, we have also highlighted the best values for each measure to indicate which method performs best for that measure.
> Then, the block structure is significant for portfolio management and our proposed framework is highly effective for portfolio analysis.
> see Table 1 on page 6, Table 3 on page 8, and Tables 4-5 on page 17.
>
> In addition, we have added the $p$-value for Beta in Table 3 on page 8, which shows that all method are significant. This, together with the lowest value of Beta for our method, implies that our method exhibits lower risk than other methods; see line 377 on page 7 and line 378 on page 8.
>
> Finally, we have included three additional methods to compared with our method, that is, the methods of
> Ledoit and Wolf (2003), Ledoit and Wolf
> (2020), and Sch{\"a}fer and Strimmer (2005).
> The results in Table 3 on page 8 indicate the additional three methods in each measure are lower than our method. Then, we can clearly find that our method significantly outperforms other methods; see lines 352-367 and 375-377 on page 7, lines 378-386 on page 8, and also see our reply to the comment \# 3 of Reviver XHvq.

---

### Official Review · Reviewer_Fohj · 2024-11-06

**Soundness:** 3
**Presentation:** 3
**Contribution:** 3
**Rating:** 6
**Confidence:** 4

**Summary:**

This paper proposes a new estimation method for the covariance matrix based on a clockwise correlation structure. The approach is straightforward: data from the same cluster are pooled and averaged as a single observation. The authors establish the asymptotic normality of the proposed estimator when cluster memberships are known for all data points. Additionally, they suggest using a spectral clustering algorithm, adapted from the network community detection literature, to identify cluster memberships. Overall, the paper is well-written, and the proposed methodology has potential applications in various fields.

**Strengths:**

The proposed method may be useful for many applications.

**Weaknesses:**

A significant theoretical gap may exist in the paper: the clustering consistency of data memberships, discussed at the end of Section 2.2, may not be sufficient to guarantee the asymptotic normality stated in Theorem 1.

**Questions:**

Although the proposed methodology is simple and straightforward, I see potential for its usefulness in certain applications. Overall, my perspective on the methodology is positive. Below are some specific questions.

1. This paper's idea is essentially simultaneous clustering and estimation of a covariance matrix. Such an idea has been studied in some recent relevant literature; see, e.g., [1], [2], [3], [4]. I think it will be helpful to discuss this line of research for a more complete picture.

2. The main theorem, Theorem 1, assumes that the group memberships for all variables are fully known. Later in the paper, however, these memberships are estimated through spectral clustering, which is shown to be consistent. I’d like to highlight a potential issue: the consistency established in Lei and Rinaldo (2015) indicates only that the percentage of mis-clustered nodes converges to zero in probability. For Theorem 1's asymptotic normality to hold, the rate of convergence would need this mis-clustered percentage to decrease faster than $n^{-1/2}$ , which appears unachievable under current assumptions. Similar challenges are noted in references [2] and [3], but these works resolve the issue by establishing almost sure convergence for group membership estimation, which may also be necessary here. Please elaborate on this issue.

3. What happens if the estimated $\hat K$ is greater than the true $K$? Intuitively, it should still be ok as long as  $\hat K$ is finite. For example, in [2] and [3], the group estimators are still consistent even if $K$ is over-specified. Can you at least provide some simulation studies to investigate this issue? This would certainly add to the applicability of the proposed methodology.

References:

[1] Su, L., Shi, Z., & Phillips, P. C. (2016). Identifying latent structures in panel data. Econometrica, 84(6), 2215-2264.

[2] Liu, R., Shang, Z., Zhang, Y., & Zhou, Q. (2020). Identification and estimation in panel models with overspecified number of groups. Journal of Econometrics, 215(2), 574-590.

[3] Zhu, X., Xu, G., & Fan, J. (2023). Simultaneous estimation and group identification for network vector autoregressive model with heterogeneous nodes. Journal of Econometrics, 105564.

[4] Liu, W., Xu, G., Fan, J., & Zhu, X. (2024). Two-way Homogeneity Pursuit for Quantile Network Vector Autoregression. arXiv preprint arXiv:2404.18732.

---

> ### Author Response · Authors · 2024-11-27
> **Reply to Question 1**
>
> Many thanks for your review. Actually, our paper's idea is a two-step estimation. First, we utilize the spectral clustering to recover the group membership $\mathbb{S}_k$s for $k=1,\cdots,K$ when the number of blocks is predetermined. Second, we estimate the unknown parameters in the blockwise correlation matrix with the estimated group membership. This is different the recent relevant literature that simultaneously estimate the model parameters and group memberships with given $K$.
> 	Those method is theoretically complex and lacking generality, since their optimization functions are non-convex and require specific algorithms.
> 	We have highlighted the difference in the revised paper; see remark 1 on page 5.

---

> ### Author Response · Authors · 2024-11-27
> **Reply to Question 2**
>
> Many thanks for your valuable review. Per your advice, we have added the almost sure convergence for the group membership estimation proved by Su et al. (2019), that is, for sufficiently large $n$ and $p$,
> $$\sup_{1\le i\le n} \sup_{1\le j\le p} \symbfit{1}_{\\{ \hat{\boldsymbol{\theta}}_j\neq \boldsymbol{\theta}_j\\}}=0, a.s.,$$
>
> see Condition (C5) on page 5.
> 	Based on this, we have obtained $$\hat{\boldsymbol{\rho}}_{\hat{\boldsymbol{\Theta}}}\stackrel{p}{\rightarrow}\hat{\boldsymbol{\rho}},$$
>
> where $\hat{\boldsymbol{\rho}}_{\hat{\boldsymbol{\Theta}}}$ and $\hat{\boldsymbol{\rho}}$ are two estimators of $\boldsymbol{\rho}$ with estimated and given group memberships, respectively.
> 	Then, we have extended the Theorem 1 to the case where group memberships are unknown; see Corollary 1 on page 5.

---

> ### Author Response · Authors · 2024-11-27
> **Reply to Question 3**
>
> Many thanks for your valuable review. According to your comment, we have conducted simulation studies with $\hat{K}=K+1$ when the group memberships are unknown and  $\boldsymbol{\epsilon}_i$ follows a multivariate normal distribution $\mathcal{N}_p(\symbfit{0}_p,\symbfit{I}_p)$.
> 	We obtain the same patterns as given $\hat{K}=K$, which implies that the blockwise correlation matrix and covariance matrix estimations are still consistent even if $K$ is over-specified;
> 	see Table R given on page 7 of the Response.pdf in Supplementary Material.
> 	In addition, the similar results are yielded when $\boldsymbol{\epsilon}_i$ follows non-normal distributions, but they are not reported here to save space.

---

### Official Review · Reviewer_VQRz · 2024-11-09

**Soundness:** 2
**Presentation:** 2
**Contribution:** 2
**Rating:** 5
**Confidence:** 3

**Summary:**

The paper proposes a method to estimate bock equicorrelation matrix, the number of blocks and recover their memberships. Theoretical results on asymptotic normality and stochastic convergence rate of parameter estimators are provided. Numerical experiments shows the effectiveness of the approach.

**Strengths:**

The paper is well-motivated and presents interesting theoretical results that justify the approach, making it applicable for high-dimensional data. The proposed method is computationally straightforward and addresses multiple aspects, including parameter estimation, block number estimation, and membership recovery.

**Weaknesses:**

1. The literature review on existing methods appears limited, which may lead to the proposed approach overlapping with established techniques. From the simple derivation below, the proposed blockwise correlation matrix estimation (BCME) in Equation (3) is closely related to Equation (12) in [1]. (Hat on $\hat{Y}$ is omitted due to compilation issue.)

$$
\frac{\frac{1}{n} \sum_{i=1}^n
  e_{p_{k_1}}^{\top} Y_{i}^{(S_{k_1})} Y_{i}^{{(S_{k_2})}^{\top}} e_{p_{k_2}} }{p_{k_1} p_{k_2}}
= \frac{
  e_{p_{k_1}}^{\top} \left( \frac{1}{n} \sum_{i=1}^n   Y_{i}^{(S_{k_1})} Y_{i}^{{(S_{k_2})}^{\top}} \right) e_{p_{k_2}} }{p_{k_1} p_{k_2}}
=  \frac{
  e_{p_{k_1}}^{\top} Cov(Y^{(S_{k_1}, S_{k_2})}) e_{p_{k_2}} }{p_{k_1} p_{k_2}}
= \frac{
 \sum_{i,j = 1}^N \rho_{ij} }{p_{k_1} p_{k_2}}
$$

Furthermore, the issue of positive definiteness mentioned in the Introduction is examined in Corollary 2 of [2].

2. The notation is not rigorous. $Y_i$ inconsistently denotes the random variable and realization (sample) of the random variable.

3. The technical conditions lack intuitive explanation, making it unclear how practical they are in real-world applications.

4. The experiments omit the turnover ratio, a significant metric in portfolio optimization analysis.

5. The topic somewhat diverges from the primary area of probabilistic methods.

References:

[1] Engle, R., & Kelly, B. (2012). Dynamic Equicorrelation. Journal of Business & Economic Statistics, 30(2), 212–228. https://doi.org/10.1080/07350015.2011.652048

[2] Archakov, I., & Hansen, P. R. (2024). A Canonical Representation of Block Matrices with Applications to Covariance and Correlation Matrices. The Review of Economics and Statistics, 106(4), 1099–1113. https://doi.org/10.1162/rest_a_01258

**Questions:**

The paper could be improved by addressing the following points

1. Providing a comprehensive review of existing methods.

2. Using more rigorous and consistent notation.

3. Offering a clearer explanation of the intuition behind the technical conditions.

4. Reporting the turnover ratio for the portfolios.

---

> ### Author Response · Authors · 2024-11-27
> **Reply to Weakness 1**
>
> We appreciate your valuable reviews. According to your advice, we have cited the two references, Engle and Kelly (2012) and Archakov and Hansen (2024), and carefully compared the differences between their estimation methods with our BCME method to clarify the contribution of our paper.
> Specifically,
> compared with the Engle and Kelly (2012, EK), we have shown the following two differences.
>
> (1a) **The different model.** Engle and Kelly (2012) imposed a blockwise structure on the Dynamic Conditional Correlation (DCC) models and proposed the Block DECO model.
>  Specifically, the Block DECO correlations are calculated as the average DCC correlation within each block (i.e., EK's equation (12)).
>  In other words, in addition to the blockwise structure, the Block DECO model incorporates additional prior structures. However, we focus solely on the blockwise correlation matrix without imposing any other structures, making our approach more general than EK's model.
>
> (1b) **The different estimation methods and theoretical assumptions.**
> With the assumptions that the variables follow a Gaussian distribution and the dimension of variables is fixed,
> Engle and Kelly (2012) employed the maximum likelihood estimation for the Block DECO model and showed the asymptotic properties of the estimated parameters.
> In contrast, we develop a novel closed-form estimator for the blockwise correlation matrix of variables using moment method (see, equation (3) in our paper). And we establish the asymptotic normality of the estimated parameters under certain moment conditions and allowing the dimension of the variables to exceed the sample size.
>
> In short, the equation (3) in our paper is the estimator for the blockwise correlation matrix and the EK's equation (12) is a DCC model with a blockwise correlation matrix, which are not closely related.
>
> Next, after carefully studying the article of Archakov and Hansen (2024, AH),
> we have found that Archakov and Hansen (2024) did examine the problem of positive definiteness mentioned in our Introduction by their Corollary 2. Then, the positive semi-definiteness of the estimated blockwise correlation matrix can simply be verified by their Corollary 2. In our paper, we similarly obey the AH's Corollary 2 and ensure the positive semi-definiteness of the estimated blockwise correlation matrix and their corresponding covariance matrix.
> In addition to this point,
> we have shown the following one difference between our paper and theirs.
>
> (1c) **The different goals.**
> The main goals of Archakov and Hansen (2024) are to derive a canonical representation for a broad class of  block matrices which includes the blockwise correlation matrices as the special cases. This canonical representation simplifies the computations of several matrix function, which improves the maximum likelihood estimation of a correlation matrix with the blockwise structure. However, our main goals are to correctly estimate the blockwise correlation matrix using moment method and establish its asymptotic properties.
>
> In addition to the differences mentioned above, we have made the following one additional contribution.
>
> (1d) **The block number determination and group membership recovery.**
> Engle and Kelly (2012) and Archakov and Hansen (2024) assumed that the block number and their memberships are given a priori. However, in real-world applications, their true values are unknown and need to be estimated correctly. We address this limitation by utilizing the ridge-type ratio criterion and spectral clustering to estimate the number of blocks and recover their memberships for a blockwise correlation matrix, and prove their consistency. Subsequently, we extend the asymptotic normality of parameter estimators and stochastic convergence rate of the estimated blockwise correlation matrix and corresponding estimated covariance matrix to the scenario where the group memberships are unknown and the block number is given.
>
> In sum, the three differences and one additional contribution mentioned above indicate our paper has considerable value; see lines 54-60 and 71-76 on page 2.

---

> ### Author Response · Authors · 2024-11-27
> **Reply to Weakness 2**
>
> Thank you for your careful reading and suggestion.
> After carefully studying the ICLR 2025 template, we have used more rigorous and consistent notation. Specifically, we have used $\mathbf{y}_i$ to represent the random variable and used $\symbfit{y}_i$ as the $i$-th realization of the random variable; see line 101 on page 2 and lines 273-274 on page 6, and also see our reply to the comment \# 1 of Reviewer KthD and comment \# 2 of Reviewer 4aHF.

---

> ### Author Response · Authors · 2024-11-27
> **Reply to Weakness 3**
>
> We apologize for causing confusion, and have made our conditions more intuitive. Per your advice, we illustrate that the Condition (C1)(i) introduces the moment conditions of $\boldsymbol{\epsilon}_i$. And we imply that Condition (C1)(ii) ensuring that the distribution does not have ''heavy tails'' (e.g., Cauchy distribution).
> Furthermore, for condition (C2), we point out that it eliminates possible multicollinearity issues; see lines 171-181 on page 4.

---

> ### Author Response · Authors · 2024-11-27
> **Reply to Weaknesses 4**
>
> Many thanks for your valuable review. We have added the turnover ratio in our paper; see Table 3 in the revised paper on page 8. Although the turnover ratio of the portfolio return based on BCME with RR is higher than that of the other portfolio return, the other measures have better performance, especially, Sharpe ratio. In short, the block structure is significant for portfolio management and our proposed framework is highly effective for portfolio analysis; see lines 367-377 on page 7 and lines 378-386 on page 8.

---

> ### Author Response · Authors · 2024-11-27
> **Reply to Weakness 5**
>
> Many thanks for your constructive reviews.
> The covariance matrix plays a fundamental role in probabilistic methods, particularly in statistical inference, parameter estimation, and model analysis. It is essential for describing the relationships, correlations, and variability between different random variables, making it a core component of many probabilistic approaches. Specifically, the covariance matrix has gained significant popularity in various fields, including but not limited to: Machine learning (Bilmes 2000; Zhang and Rao 2013), finance and risk management (Markowitz 1952; Jagannathan and Ma 2003), econometrics (Chen and Conley 2001; Fan et al. 2008), biostatistics (Tong and Wang 2007; Friedman et al. 2008), and neuroscience and gene expression (Parketal. 2007; Wu and Smyth 2012; Tan et al. 2015; Eisenach et al. 2020; Pircalabelu and Claeskens 2020).

---

> > ### Comment · Reviewer_VQRz · 2024-11-27
> >
> > I appreciate the authors' efforts in providing clarification. However, the authors did not address the mathematical derivation, which suggests that the closed-form solution is already known. Additionally, the overly short backtesting period raises concerns about potential cherry-picking. The higher TR and statistically insignificant improvement in SR also do not inspire confidence in the strategy's effectiveness. For these reasons, I will maintain my original score.

---

### Meta-Review · Area_Chair_WL4C · 2024-12-19

**Metareview:**

The paper proposes a closed-form estimator for correlation matrix with blockwise structure, together with a method to estimate the number of blocks and recover their memberships. Theoretical results on asymptotic normality and stochastic convergence rate are provided, along with numerical experiments.

Reviewers generally agree that the theoretical results are interesting and justify the presented estimators. However, there are concerns about the novelty of the estimator and the soundness of the analysis.

**Additional Comments On Reviewer Discussion:**

- Reviewer VQRz: pointed out the that Eq(3) is closely related to Eq(12) in Engle and Kelly (2012), suggesting that the closed form estimator is not new. The authors did not address the mathematical derivation discussed by the reviewer.

- Reviewer Fohj: A significant theoretical gap may exist in the paper: the clustering consistency of data memberships may not be sufficient to guarantee the asymptotic normality stated in Theorem 1. The authors responded by assuming the almost sure convergence of group membership (proved by Su et al 2019 under additional assumptions) and state Corollary 1. The proof of Corollary 1 is not given, however.

- Reviewer Fohj: if the estimated $\hat{K}$ is greater than the true $K$, the reviewer pointed out that the group estimators in Liu et al (2020) and Zhu et al (2023) are still consistent and asked for at least some simulation studies. The authors carried out the simulation with $\hat{K} = K+1$ when $\epsilon_i$ is under the normal distribution. I believe that either a theoretical study or more simulations would be needed to address this point.

These points, among others, lead to the reject decision of the paper.

---

### Decision · Program_Chairs · 2025-01-22

Reject